# Activation of Nkx2.5 transcriptional program is required for adult myocardial repair

Carmen de Sena-Tomás [1], Angelika G. Aleman[2], Caitlin Ford[3], Akriti Varshney[4,5], Di Yao[1], Jamie K. Harrington[1], Leonor Saúde [6], Mirana Ramialison [5,7] & Kimara L. Targoff [1,8✉]

The cardiac developmental network has been associated with myocardial regenerative potential. However, the embryonic signals triggered following injury have yet to be fully elucidated. Nkx2.5 is a key causative transcription factor associated with human congenital heart disease and one of the earliest markers of cardiac progenitors, thus it serves as a promising candidate. Here, we show that cardiac-specific RNA-sequencing studies reveal a disrupted embryonic transcriptional profile in the adult Nkx2.5 loss-of-function myocardium. $nkx2.5^{-/-}$ fish exhibit an impaired ability to recover following ventricular apex amputation with diminished dedifferentiation and proliferation. Complex network analyses illuminate that Nkx2.5 is required to provoke proteolytic pathways necessary for sarcomere disassembly and to mount a proliferative response for cardiomyocyte renewal. Moreover, Nkx2.5 targets embedded in these distinct gene regulatory modules coordinate appropriate, multi-faceted injury responses. Altogether, our findings support a previously unrecognized, Nkx2.5-dependent regenerative circuit that invokes myocardial cell cycle re-entry, proteolysis, and mitochondrial metabolism to ensure effective regeneration in the teleost heart.

[1] Division of Cardiology, Department of Pediatrics, College of Physicians & Surgeons, Columbia University, New York, NY 10032, USA. [2] Department of Physiology & Cellular Biophysics, College of Physicians & Surgeons, Columbia University, New York, NY 10032, USA. [3] Department of Genetics & Development, College of Physicians & Surgeons, Columbia University, New York, NY 10032, USA. [4] Monash Biomedicine Discovery Institute, Monash University, Clayton, VIC 3800, Australia. [5] Australian Regenerative Medicine Institute & Systems Biology Institute Australia, Monash University, Clayton, VIC 3800, Australia. [6] Instituto de Medicina Molecular, Faculdade de Medicina da Universidade de Lisboa, 1649-028 Lisboa, Portugal. [7] Murdoch Children's Research Institute & Department of Peadiatrics, The University of Melbourne, Parkville, VIC 3052, Australia. [8] Columbia Stem Cell Initiative, Columbia University, New York, NY 10032, USA. ✉email: kl284@columbia.edu

D ue to the limited responsiveness of endogenous cellular and molecular repair mechanisms in injured myocardium, cardiac dysfunction remains a key cause of morbidity and mortality in patients. Coronary artery disease leads to myocardial infarction which results in fibrotic scarring, adverse remodeling, and heart failure[1]. Furthermore, in patients with congenital heart disease, cardiac ischemia yields cardiomyocyte (CM) loss and the induction of inappropriate healing responses[2]. Despite these pathogenic mechanisms that ensue following injury, recent reports demonstrate that adult CMs preserve the capacity to replicate in both mice[3–8] and humans[9–11]. Therefore, identifying key factors that enhance CM proliferation offers opportunities to stimulate myocardial repair in patients suffering from cardiac dysfunction or hypoplasia.

Interestingly, in contrast to adult mice and humans, neonatal mice and adult zebrafish retain the ability to undergo CM dedifferentiation and proliferation, optimizing cardiac regenerative potential[12–15]. Specifically, new CMs emerge following division of pre-existing CMs to repopulate the injury site[12,13]. Furthermore, despite the propensity of most mammalian CMs to undergo polyploidization[16], a positive correlation between the maintenance of a mononuclear, diploid state, and retention of cardiac regeneration capacity in both mice and zebrafish has been established[17,18]. Taken together, exploiting the pro-regenerative mechanisms innate in zebrafish and applying them to diseased human hearts would provide an arsenal of tools to awaken constructive healing in both adult and pediatric patients[19,20].

Parallel transcriptional networks have been shown to be essential for both development and regeneration of several tissues including the heart[13,21,22]. For example, reactivation of genes essential during cardiogenesis is critical to stimulate CM production in damaged myocardium[23]. Thus, we took advantage of our prior investigation of the Nkx2.5 transcriptional regulatory pathways in the zebrafish embryo to provide insights into the developmental mechanisms that are harnessed for activation of cardiac regenerative potential. The homeodomain transcription factor, NKX2-5, is one of the most commonly mutated genes associated with human congenital heart disease (CHD) and is a master regulator of cardiac development[24]. Studies in model organisms highlight crucial roles of vertebrate and invertebrate homologs of NKX2-5 in cardiac specification and morphogenesis[25–28]. While the functions of Nkx genes in early development have been examined, it remains unclear whether Nkx2-5 is necessary to mount an efficient response for CM renewal. In this study, we inspect the distinct cellular and molecular processes regulated by Nkx2-5 in injured myocardium to elucidate effective therapeutic strategies for myocardial regeneration.

Our previous work illuminates that temporally controlled expression of nkx2.5 is required to preserve chamber-specific identity maintenance in both the first and second heart fields during discrete developmental time windows[29–31]. Moreover, in the context of otherwise embryonic lethal nkx2.5−/− embryos, early overexpression (21 somites) of nkx2.5 is sufficient to sustain cardiac function and ensure embryonic viability into adulthood[29]. These results regarding the temporal regulation of nkx2.5 broaden our appreciation of the essential functions of early cardiac transcriptional regulation in establishing long-standing myocardial health. Thus, we exploited this unique model to study the role of nkx2.5 in the adult heart following injury. Here, we show that rescued nkx2.5−/− fish demonstrate a diminished regenerative response following amputation of the ventricular apex. Compromised sarcomere dedifferentiation and decreased proliferation dampen the requisite regrowth of new CMs from pre-existing CMs in the nkx2.5−/− fish. Moreover, the epicardium fails to penetrate the regenerate to establish the tissue environment necessary to induce a cascade of signals required for

healing. Altogether, our data illustrate the critical function of Nkx2.5 in the stressed adult myocardium to invoke regenerative cues for CM dedifferentiation, renewal, and effective patterning of a supportive microenvironment. Understanding the mechanisms guiding Nkx2.5-dependent cardiac regeneration will have an impact on the management of patients with CHD or cardiomyopathies associated with NKX2-5 mutations.

## Results

**Embryonic rescue generates nkx2.5 loss-of-function fish.** Given the embryonic lethality of mutations in nkx2.5[31], we pioneered an inducible, overexpression reagent to study the role of nkx2.5 in the adult myocardium. In nkx2.5−/− embryos, heat shock of Tg(hsp70l:nkx2.5-EGFP) during heart tube formation (21 somites) rescues the cardiac chamber proportion and identity defects normally observed during embryogenesis (Fig. 1A)[29]. These results led us to investigate the gross morphology and chamber-specific characteristics in age- and size-matched, heat-shocked, non-transgenic wild-type, Tg(hsp70l:nkx2.5-EGFP), and nkx2.5−/−;Tg(hsp70l:nkx2.5-EGFP) (hereafter referred to as nkx2.5−/−) fish (Fig. 1). Remarkably, the myocardial architecture appears normal in adult nkx2.5−/− fish (MT0) as compared to non-transgenic (WT0) and transgenic (TG0) wild-type fish following heat shock at 21 somites (Fig. 1B–D). Employing Acid Fuchsin-Orange G (AFOG), there is no evidence of tissue damage or collagen deposition in non-transgenic wild-type, transgenic wild-type, and nkx2.5−/− fish (Fig. 1E–G). Tropomyosin immunofluorescence delineates clearly ordered and uniformly spaced z-disks in all three genotypes (Fig. 1N–P). Furthermore, ventricular myosin heavy chain (vmhc) is expressed uniformly throughout the ventricles of the non-transgenic wild-type, transgenic wild-type, and nkx2.5−/− fish (Fig. 1H–J). However, while atrial myosin heavy chain (amhc) expression is visualized appropriately in the atrium in all genotypes, it is uniquely expressed in the ventricle of the nkx2.5−/− fish (Fig. 1M), a finding that is reminiscent of ectopic atrial (S46+) CMs in nkx2.5−/− embryos[29]. Although these results reinforce the temporal requirement for Nkx2.5 in ventricular identity maintenance[29], we observed no evidence of gross histological or structural malformations of the ventricle or atrium in the nkx2.5−/− fish.

To substantiate our findings with quantitative strategies, we applied previously established morphometrics to assess for normal postembryonic cardiac growth. Precisely, we observed no statistically significant difference in the percentage of trabeculation between non-transgenic, transgenic wild-type, and nkx2.5−/− hearts (Fig. 1Q)[32]. However, ventricular volume is slightly increased in the nkx2.5−/− fish when compared to non-transgenic and transgenic wild-type fish, suggesting mild ventricular expansion (Fig. 1R). To assess the significance of this subtle variability in chamber size, we tested for possible signs of cardiac failure by subjecting the fish to an exercise routine[33]. In a swim tunnel, we measured endurance, the average time during which animals could maintain orientation facing a high velocity current, and found no statistically significant differences between all genotypes assessed (Fig. 1S). Together, we conclude that, although future investigation may help to unravel these slight differences in ventricular size, Nkx2.5 is not necessary following embryogenesis to ensure long-term viability and fitness into adulthood.

**Molecular signatures of adult nkx2.5−/− hearts are perturbed.** Despite intact cardiac formation in the nkx2.5−/− compared to wild-type fish, we posited that there may be alterations in the underlying molecular signatures given the absence of this critical cardiac transcription factor. Thus, we performed RNA sequencing (RNA-seq) analysis of ventricular apices extracted from non-transgenic wild-type, transgenic wild-type, and nkx2.5−/− hearts.

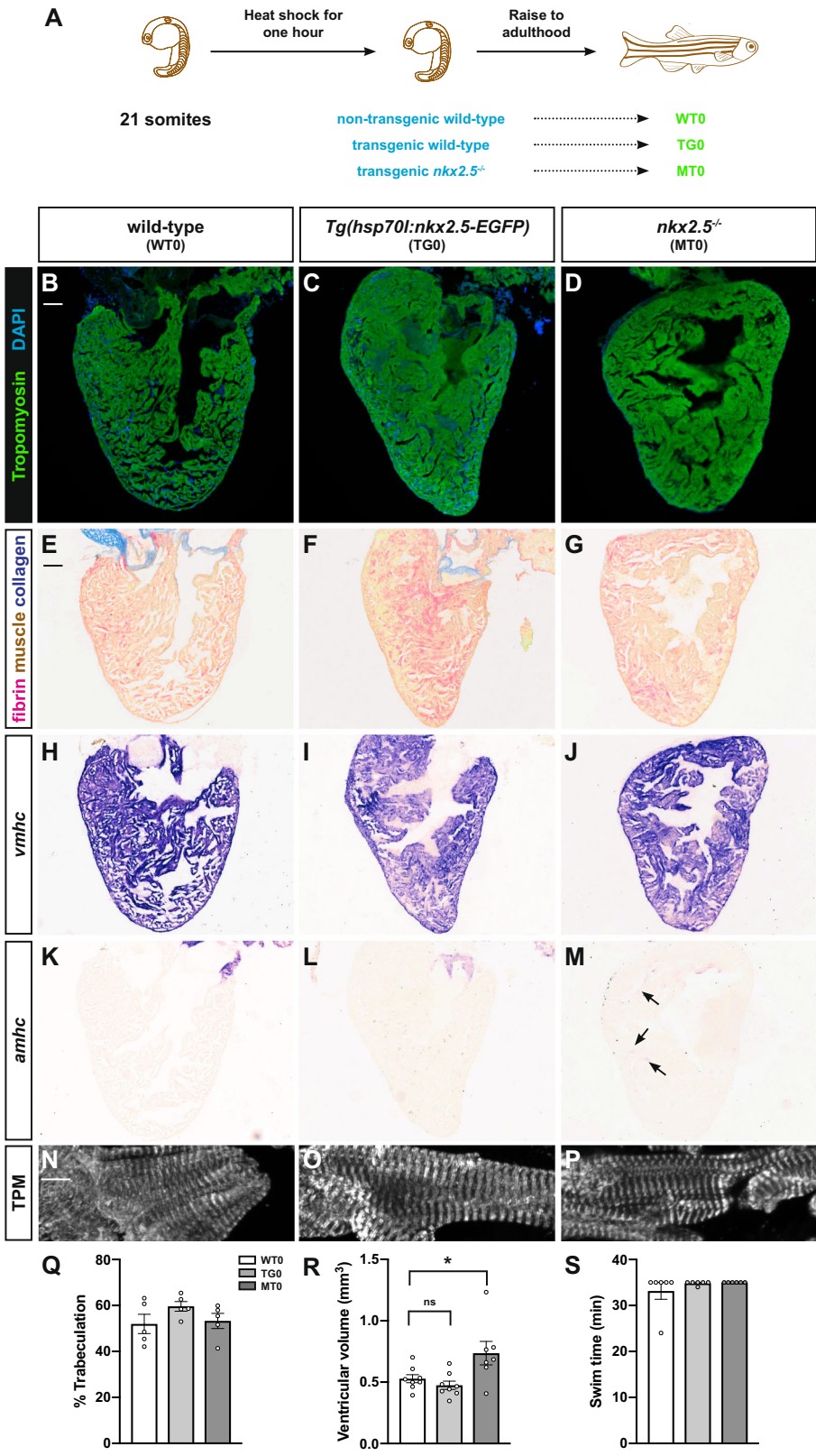

We investigated the fundamental molecular similarities and differences of these three genotypes by retrieving the subsets of statistically significant, differentially expressed genes (DEGs). To isolate the set of DEGs that are misregulated in the $nkx2.5^{-/-}$ fish (MT0) independent of the control sample used (WT0 or TG0), we established a subgroup of 1073 genes (hereafter referred to as group C or the 'overlap' group, Fig. S1A). This intersection category reflects the critical genes that are differentially expressed in the $nkx2.5^{-/-}$ heart when compared to both wild-type conditions while excluding those genes that are misregulated between the transgenic and non-transgenic wild-type samples.

We then explored the Gene Ontology (GO) categories most highly enriched in each of the comparisons. Although several GO terms were similarly enriched in the MT0 vs WT0 (group A) and

**Fig. 1 Adult nkx2.5⁻/⁻ fish have structurally normal hearts. A** Schematic of the experimental design and genotype identification. **B–D** Representative images of hearts stained for Tropomyosin (green) and DAPI (blue) indicate cardiac muscle and nuclei, respectively, in wild-type (WT0) ($n = 11$) (**B**), *Tg(hsp70l:nkx2.5-EGFP)* (TG0) ($n = 6$) (**C**), and *nkx2.5⁻/⁻* (MT0) ($n = 7$) (**D**) adult hearts. Scale bar, 100 μm. **E–G** The same sections in the top panels are stained with Acid Fuchsin-Orange G (AFOG) to indicate fibrin (red), myocardium (orange), and collagen (blue) in non-transgenic wild-type (WT0) ($n = 11$) (**E**), transgenic wild-type (TG0) ($n = 6$) (**F**), and *nkx2.5⁻/⁻* (MT0) ($n = 7$) (**G**) adult hearts. Scale bar, 100 μm. **H–M** In situ hybridization for *vmhc* (**H–J**) and *amhc* (**K–M**) in non-transgenic wild-type (WT0) ($n = 11$) (**H, K**), transgenic wild-type (TG0) ($n = 6$) (**I, L**), and *nkx2.5⁻/⁻* (MT0) ($n = 7$) (**J, M**) adult hearts. In *nkx2.5⁻/⁻* hearts, *amhc* is expressed ectopically in the ventricle (arrows) ($n = 7/7$). **N–P** Tropomyosin immunostaining elucidates well-organized z-disks in non-transgenic wild-type (WT0) ($n = 11$) (**N**), transgenic wild-type (TG0) ($n = 6$) (**O**), and *nkx2.5⁻/⁻* (MT0) ($n = 7$) (**P**) ventricular myocardium. Scale bar, 10 μm. **Q, R** Morphometric evaluation of trabecular myocardium in non-transgenic wild-type (WT0) ($n = 5$), transgenic wild-type (TG0) ($n = 5$), and *nkx2.5⁻/⁻* (MT0) ($n = 5$) adult hearts and ventricular volume in non-transgenic wild-type (WT0) ($n = 8$), transgenic wild-type (TG0) ($n = 8$), and *nkx2.5⁻/⁻* (MT0) ($n = 7$) adult hearts. Mean and standard error of each data set are shown. Unpaired, two-tailed *t*-test demonstrates no statistically significant differences between WT0 and TG0 ($p = 0.1428$) and WT0 and MT0 ($p = 0.8124$) in **Q**. Unpaired, two-tailed *t*-test illustrates no statistically significant difference between WT0 and TG0 ($p = 0.2770$) and statistically significant difference between WT0 and MT0 ($p = 0.0490$) in **R**. **S** Swim endurance assessed in non-transgenic wild-type (WT0) ($n = 6$), transgenic wild-type (TG0) ($n = 6$), and *nkx2.5⁻/⁻* (MT0) ($n = 6$) fish at maximal water speed. Mean and standard error of each data set are shown. Unpaired, two-tailed *t*-test shows no statistically significant differences between WT0 and TG0 ($p = 0.3866$) and WT0 and MT0 ($p = 0.3409$). Source data are provided as a Source Data file.

the MT0 vs TG0 (group B) assessments (Fig. 2A), we refer to the 'overlap' group as the most accurate depiction of the DEGs between the adult *nkx2.5⁻/⁻* and wild-type hearts. Muscle structure development, actin filament-based processes, heart development, and circulatory system processes are the enhanced GO terms (Fig. 2A, 'overlap' group), underlining our findings that the developmental pathways regulated by Nkx2.5 in the embryo are analogously disrupted in the adult loss-of-function model. We also generated a heat map from the 'overlap' group that displays the z-score for genes with relatively higher or lower expression levels between samples (Fig. 2F). This representation of the 'overlap' dataset underscores the consistency between the replicates within each genotype. Moreover, these data emphasize that approximately 2/3 of the DEGs in the *nkx2.5⁻/⁻* fish are upregulated and 1/3 is downregulated, consistent with reported repressive functions of Nkx2.5[34–36].

While this methodical strategy of defining the 'overlap' category captures variations in the molecular profile unique to the loss of *nkx2.5* gene function, we questioned whether *hsp70l* promotor leakiness could explain the observed differences between TG0 and WT0. Our prior data illustrates absence of *Tg(hsp70l:nkx2.5-EGFP)* expression during embryogenesis[29]. We thus inspected the potential for overexpression in the adult heart. Interestingly, qPCR analysis of *nkx2.5* expression illuminates no statistically significant difference between levels detected in TG0 and WT0 hearts (Fig. S1B). Given that the presence of *Tg(hsp70l:nkx2.5-EGFP)* does not explain the differences observed between TG0 and WT0 datasets, we dissected the most highly enriched GO terms in group D (Fig. 2A) and found autophagy, mitochondrion organization, and ribosome biogenesis. Thus, we conclude that aberrations in these 'housekeeping' functions constitute the primary differences in this comparison and are distinct from the molecular changes observed in the MT0 comparisons. Indeed, while 'housekeeping' functions represent >75% of the DEGs in the TG0 vs WT0 comparison, these GO terms compose only ~5% of the DEGs in the MT0 comparisons, reinforcing the benefits of utilizing the 'overlap' dataset to isolate genetic differences attributed to the loss of *nkx2.5* gene function in the adult heart.

In this 'overlap' set, we investigated the genetic interaction network pertaining to the heart development GO category. Our studies identify *bmp4*, a vital signal in cardiac development and regeneration, as a core node (Fig. 2B). *bmp4* is upregulated in the *nkx2.5⁻/⁻* fish when compared to both non-transgenic and transgenic wild-type fish. Our results are closely aligned with previous murine studies in which a negative feedback loop is identified between Nkx2-5 and Bmp2[25], the functional equivalent

of zebrafish Bmp4[37]. Given that *bmp4* serves as a central connector in our RNA-seq dataset, we propose that modulation of myriad Nkx2.5 effectors, both direct and indirect, function to disrupt the cardiac development regulatory program in the *nkx2.5⁻/⁻* fish. As verification of the biological significance of this gene regulatory network (GRN), we detected statistically significant upregulation of versican (*vcana*) in *nkx2.5⁻/⁻* compared to the wild-type myocardium employing RNAscope strategy (Fig. 2C–E). Taken together, our results illuminate that overexpression of Nkx2.5 at 21 somites rescues *nkx2.5⁻/⁻*-associated embryonic defects but results in enduring perturbation of the essential developmental pathways throughout adulthood.

**Activation of *nkx2.5* occurs following myocardial injury.** Given that the Nkx2.5 downstream network is modified despite normal myocardial infrastructure in *nkx2.5⁻/⁻* hearts, we hypothesized that the integrity of the Nkx2.5 transcriptional profile is robust enough to obfuscate gross structural abnormalities. We speculated that stress or injury would expose a more dramatic disruption in the molecular program transcriptionally regulated by Nkx2.5 in the adult heart. Yet, the specific nature of *nkx2.5* expression in the mature wild-type zebrafish heart upon injury has yet to be fully elucidated. Previous studies have reported expression of *nkx2.5* along the resection plane[13,23], while others have been unable to detect *nkx2.5* transcripts through in situ hybridization (ISH)[12]. Thus, we performed ventricular resections in wild-type fish, as previously described[38]. Using RNAscope, our data reveal expression of *nkx2.5* throughout the ventricular myocardium in uninjured and at 7 days and 14 days post amputation (dpa) (Fig. 3A–C). Moreover, upon examination of the injured area, it is evident that a few cells are also expressing *nkx2.5* inside the regenerate (Fig. 3B, C). Intriguingly, at 7 dpa, minimal *nkx2.5* expression is observed in the wound area. However, the abundance of transcripts increases progressively over time, a pattern that reflects the process of redifferentiation of newly generated CMs. We repeated the injury in wild-type fish carrying *Tg(nkx2.5:ZsYellow)* and detected both endogenous ZsYellow and Tropomyosin through immunostaining. Our RNAscope findings were reinforced by co-expression of ZsYellow and Tropomyosin in both uninjured and injured hearts (Fig. 3D–F'). Together, these data substantiate our conclusion that Nkx2.5 is expressed in the adult zebrafish myocardium and in the regenerate following ventricular amputation.

Our next goal was to clarify the lineage-specific expression patterns of *nkx2.5* in the adult heart. To delineate endothelial and epicardial cells from CMs, we performed immunostaining with *Tg(nkx2.5:ZsYellow)* in conjunction with Raldh2 antibody[39]. At

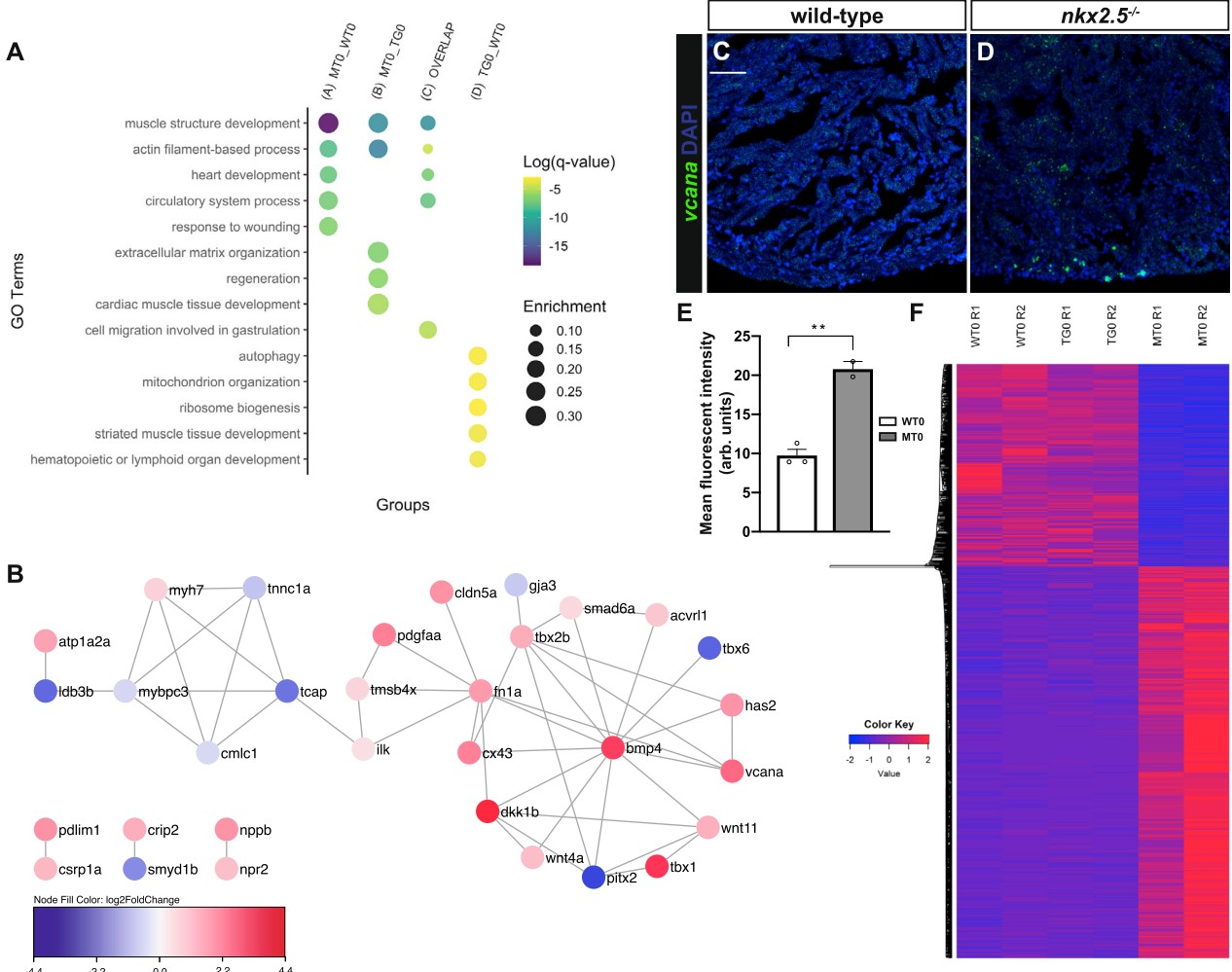

**Fig. 2 Developmental transcriptional pathways are disrupted in the adult *nkx2.5*^−/− myocardium. A** Gene Ontology (GO) analysis of all the DEGs in groups A, B, C, D from Fig. S1. The five most significantly enriched GO terms are selected for each comparison. Color and size of circles correspond to log (q-value) and enrichment, respectively. **B** Network analysis of genes associated with the enriched GO term, heart development, from the 'overlap' set (group C). Color represents the fold change comparisons between *nkx2.5*^−/− and wild-type hearts (MT0 vs WT0). Red is upregulated and blue is downregulated, as shown in the key. **C**, **D** Representative images of RNAscope in situ hybridization analysis on sections from uninjured hearts employing a probe directed against *vcana* highlight upregulation in *nkx2.5*^−/− compared to wild-type myocardium. Scale bar, 50 μm. **E** Quantification of integrated signal density of *vcana* is depicted in wild-type ($n = 3$) and *nkx2.5*^−/− ($n = 2$) hearts. Mean and standard error of each data set are shown. Unpaired, two-tailed *t*-test yields a statistically significant difference between WT0 and MT0 ($p = 0.0031$). **F** Hierarchical clustering heatmap analysis of DEGs ($n = 1071$) noted in the 'overlap' set (group C). The *Z*-scores of gene expression measurements are displayed as colors ranging from blue to red. The rows are clustered using Euclidean distance measures and average linkage. Source data are provided as a Source Data file.

7 dpa, ZsYellow+ cells do not co-localize with Raldh2+ cells suggesting absence of *nkx2.5* expression in the endothelial and epicardial lineages (Fig. 3G–G‴). Furthermore, we used vimentin to label fibroblasts in *Tg(nkx2.5:ZsYellow)* fish and identified no evidence of overlapping expression between endogenous ZsYellow+ signal in CMs and Vimentin+ fibroblasts (Fig. 3H–H‴)[40]. From these data, we confirm that expression of Nkx2.5 is limited to the CM population in the adult heart.

**Nkx2.5 is required for myocardial regeneration.** Confirmation of *nkx2.5* expression in the amputation plane of the adult zebrafish myocardium provided further ammunition to our hypothesis that loss of Nkx2.5 transcriptional regulation in the context of injury would intensify the disruption in the downstream molecular program. To address this question, we performed ventricular resections in genotyped *nkx2.5*^−/− fish. We selected this injury tactic to circumvent inadvertent activation of

the *hsp70l* transgene given previous studies documenting transgenic expression for as long as six days following cryoinjury[41]. To verify that the physical trauma of amputation does not yield a similar effect, we measured the magnitude and duration of *nkx2.5* transcriptional activation in *Tg(hsp70l:nkx2.5-EGFP)* fish following ventricular resection, but in the absence of heat shock. Executing qPCR, we observed peak expression 1hour post amputation and normalization to endogenous levels by 2 dpa (Fig. S2F). Moreover, employing an anti-GFP antibody, we implemented immunostaining to detect the spatiotemporal expression of Nkx2.5-EGFP following ventricular amputation. Our results uncover parallel, albeit slightly delayed in comparison to qPCR evaluation of *nkx2.5-EGFP* transcripts (Fig. S2F), expression of Nkx2.5-EGFP with diminution of the protein by 2 dpa (Fig. S2A–E). Nkx2.5-EGFP is primarily evident within ventricular myocardium and not specifically at the injury border. Moreover, Nkx2.5-EGFP expression is cytoplasmic initially

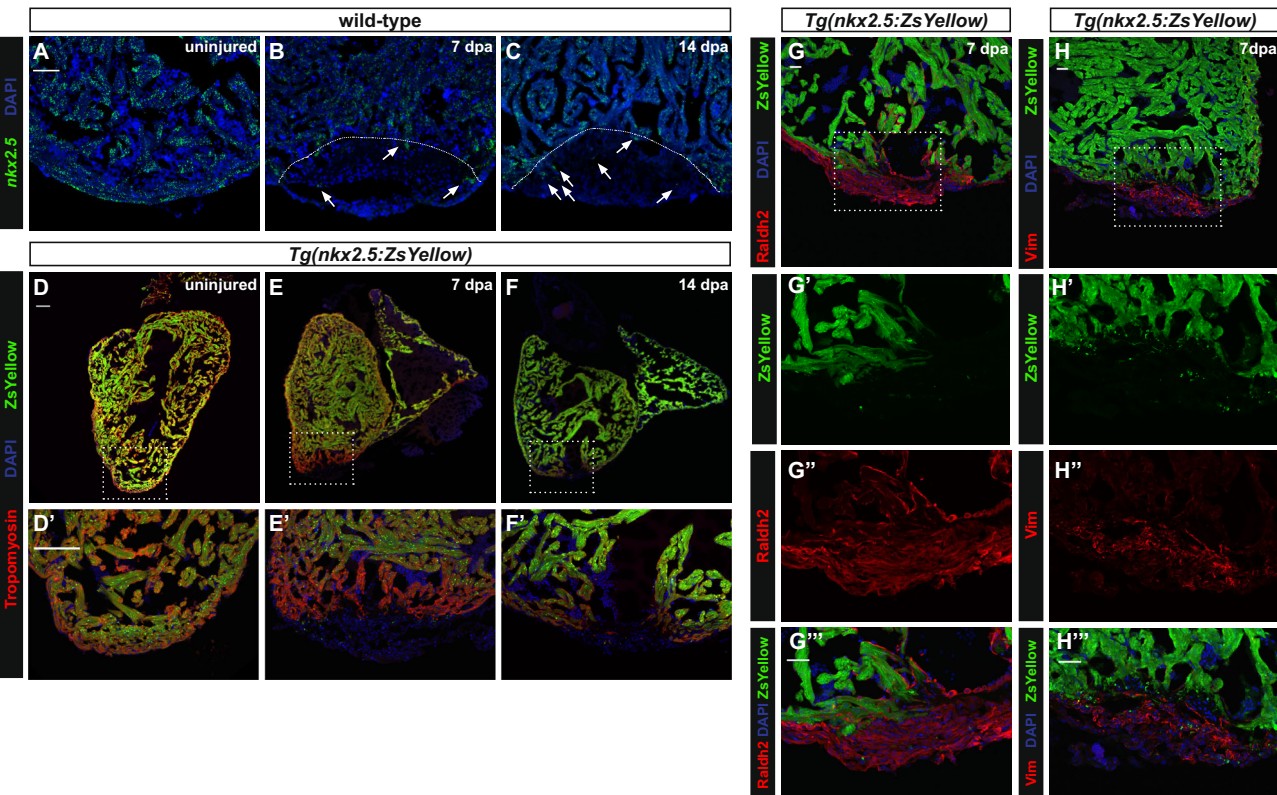

**Fig. 3 Adult zebrafish cardiomyocytes activate *nkx2.5* expression following injury. A–C** Representative images of RNAscope in situ hybridization on sections of uninjured (*n* = 3) (**A**) and injured wild-type hearts at 7 dpa (*n* = 3) (**B**), and 14 dpa (*n* = 3) (**C**) highlight *nkx2.5* expression in the entire ventricle, expanding into the regenerate. Dashed lines represent amputation planes. Scale bar, 50 μm. **D–F′** Co-localization of *Tg(nkx2.5:ZsYellow)* and Tropomyosin delineates *nkx2.5* expression in the myocardium in both uninjured (*n* = 4) (**D**) and injured wild-type hearts at 7 dpa (*n* = 5) (**E**) and 14 dpa (*n* = 6) (**F**). Higher magnification panels (**D′–F′**) indicate the areas outlined in the rectangular boxes. Scale bar, 100 μm. **G–H‴**) Anti-Raldh2 (*n* = 4) (**G–G‴**) and anti-Vimentin (*n* = 5) (**H–H‴**) immunostainings in *Tg(nkx2.5:ZsYellow)* fish highlight the myocardial-specific expression of *nkx2.5*. Higher magnification panels (**G′–G‴**, **H′–H‴**) indicate the areas outlined in the rectangular boxes. Scale bar, 50 μm.

(Fig. S2A–C) and becomes localized to the nucleus at 1 dpa (Fig. S2D) prior to exhibiting downregulation at 2 dpa (Fig. S2E). Taken together, these data suggest that, while *Tg(hsp70l:nkx2.5-EGFP)* is activated in response to ventricular resection, *nkx2.5* expression is limited in duration and location, thus, minimizing the likelihood that these secondary effects modulate the response to injury.

To test our hypothesis that Nkx2.5 is required for adult myocardial repair, we exploited this ventricular apex amputation approach to injure wild-type and *nkx2.5*$^{−/−}$ fish (Fig. 4A). We were fascinated to detect diminished regenerative potential with increased fibrin retention and deposition of collagen at the injury site at 30 dpa in *nkx2.5*$^{−/−}$ compared to non-transgenic and transgenic wild-type hearts (Figs. 4B–D′, S3). Quantification of the percentage of scar tissue in relation to each ventricle (Fig. 4E) and the degree of residual scar (Fig. 4F) at 30 dpa substantiates these qualitative findings. Moreover, at 50 dpa, the pronounced defect in muscularization is still evident in the *nkx2.5*$^{−/−}$ compared to the non-transgenic and transgenic wild-type hearts (Fig. S4A–L′) with statistically significant differences manifest by residual scar tissue and severity analyses (Fig. S4M, N). In addition to evaluating the repair mechanisms in the *nkx2.5*$^{−/−}$ fish, we also examined their chamber identity maintenance. Although the ventricular myocardium in *nkx2.5*$^{−/−}$ fish exhibits areas of ectopic *amhc* expression (Fig. 1M, S5C), the healing injury site preserves its chamber-specific characteristics, similar to non-transgenic and transgenic wild-type hearts (Fig. S5A, B). Our findings suggest that, while *nkx2.5* is required for embryonic[31]

and adult ventricular identity maintenance (Fig. 1M, S5C), other transcriptional regulatory mechanisms are essential to prevent transdifferentiation of ventricular myocardium in the context of CM renewal. Altogether, these experiments reveal an essential requirement for Nkx2.5 to induce effective myocardial regeneration in the adult teleost heart.

Given our findings that the molecular signature of adult *nkx2.5* loss-of-function myocardium exhibits disruption in the normal developmental networks without evidence of detectable cardiac structural abnormalities (Figs. 1B–G and 2B), we questioned whether these disrupted pathways are required for successful stress response or injury repair. Thus, we performed a transcriptomic profiling from non-transgenic wild-type (WT7), transgenic wild-type (TG7), and *nkx2.5*$^{−/−}$ (MT7) fish at 7 days dpa. First, we investigated the intersection of the regenerative responses in both control systems, TG7 vs TG0 and WT7 vs WT0. From this approach, we identified 882 and 408 DEGs that are commonly upregulated and downregulated, respectively (Fig. S6A, B, green). Applying GO enrichment for these shared DEGs, we affirmed that the normal cardiac regenerative transcriptional profile is recruited as illustrated by highly enriched categories such as extracellular matrix organization, actin cytoskeleton organization, and mitotic sister chromatid segregation (Fig. S6C). Furthermore, we validated the RNA-seq expression profile of essential genes known to be imperative for repair mechanisms using qPCR. For the WT7 vs WT0 comparison, we focused on 14 genes derived from the overlapping dataset representing the shared DEGs in the RNA-seq

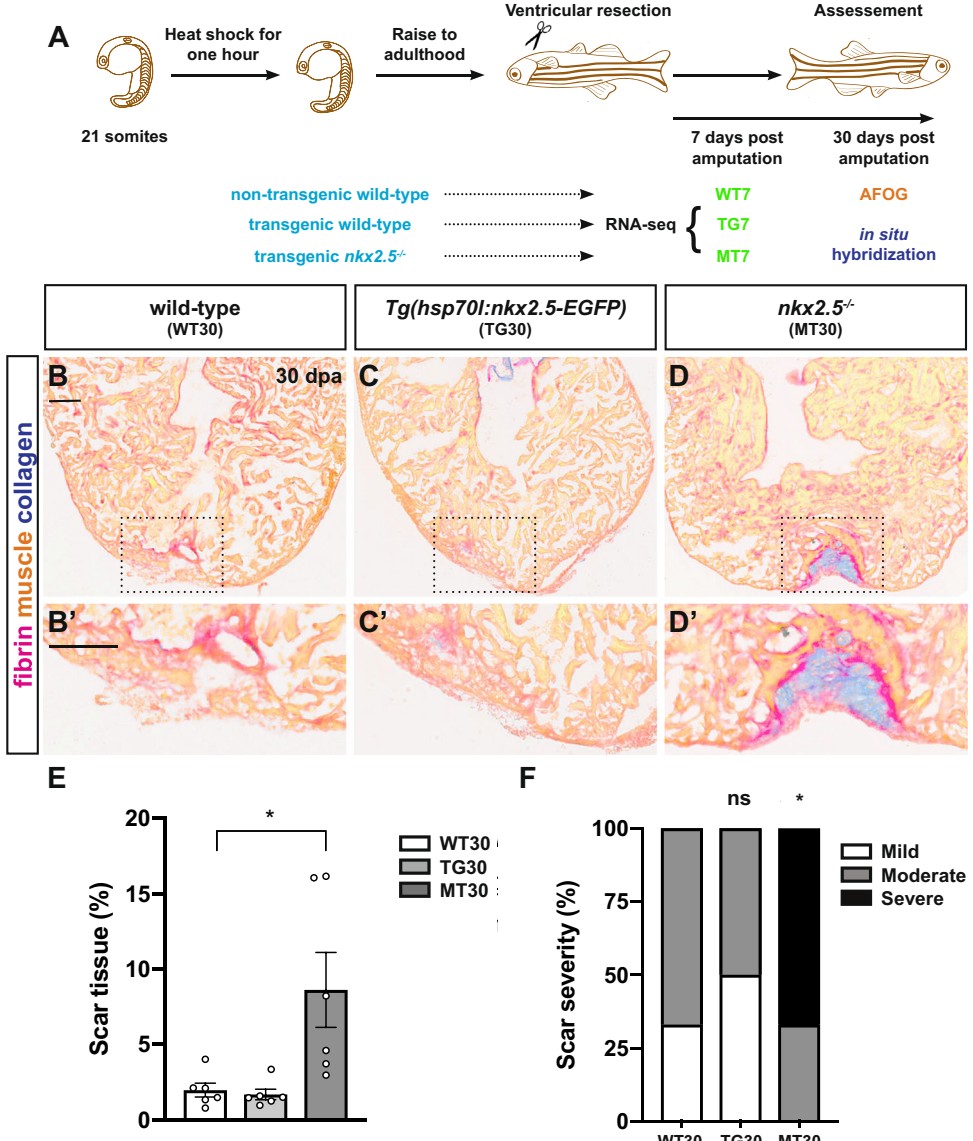

**Fig. 4 Nkx2.5 is required for myocardial regeneration. A** Schematic of the experimental design. **B–D** Impaired healing is evident in the *nkx2.5$^{-/-}$* hearts (*n* = 6) (**D**) when compared with the non-transgenic wild-type (*n* = 6) (**B**) and transgenic wild-type (*n* = 6) (**C**) AFOG-stained adult hearts at 30 days post-amputation (dpa). Scale bar, 100 μm. **B'–D'** Higher magnification images of the rectangular boxes in **B–D**. Scale bar, 100 μm. **E, F** Quantification of the percentage of scar tissue per ventricle (**E**) and the degree of scar severity (**F**) in non-transgenic wild-type (*n* = 6), transgenic wild-type (*n* = 6), and *nkx2.5$^{-/-}$* (*n* = 6) fish illustrate the dramatically compromised injury response in the absence of *nkx2.5* gene function. Mean and standard error of each data set are shown. Unpaired, two-tailed *t*-test demonstrates no statistically significant difference between WT30 and TG30 (*p* = 0.6065), but statistically significant difference between WT30 and MT30 (*p* = 0.0251) in **E**. Chi-squared test shows no statistically significant difference between WT30 and TG30 (*p* = 0.5582) and statistically significant difference between WT30 and MT30 (*p* = 0.0357). Source data are provided as a Source Data file.

control systems (Fig. S6A, B, green). While only two genes reached statistical significance globally, we ascertained that 11 of the 14 transcripts were similarly up or downregulated by qPCR, corroborating the accuracy of our wild-type sequencing data (Fig. S6D, E). Finally, PCA comparing all RNA-seq samples shows that duplicates for each condition cluster together, indicating that the DEGs are not due to noise (Fig. S6F). Second, the first dimension tends to separate injured versus uninjured samples, while the second dimension tends to separate samples by genotype (WT, TG, and MT). In summary, these data indicate robust, global transcriptional differences in MT versus WT or TG samples. However, we observed a larger separation between the WT0 and TG0 samples, which we have resolved to be secondary to differences in 'housekeeping' functions (Fig. S1C), and account

for this variation in our analysis by removing the contribution of these DEGs to the WT0 and TG0 comparison by generating the 'overlap' set (Fig. 2A, group C). Together, we show that the intersection of the regenerative responses in both control systems reflects previously identified cellular repair processes and that our RNA-seq transcriptional profiles echo in vivo alterations in transcriptional regulation.

**Nkx2.5 deficiency disrupts proliferation and proteolysis.** Following validation of the endogenous reparative mechanisms in our control fish after amputation (Fig. S6), we aimed to probe the underlying causes of the diminished regenerative potential in the *nkx2.5$^{-/-}$* fish at 7 dpa. To accomplish this goal, we generated an UpSet plot to extract the specific gene sets that authentically

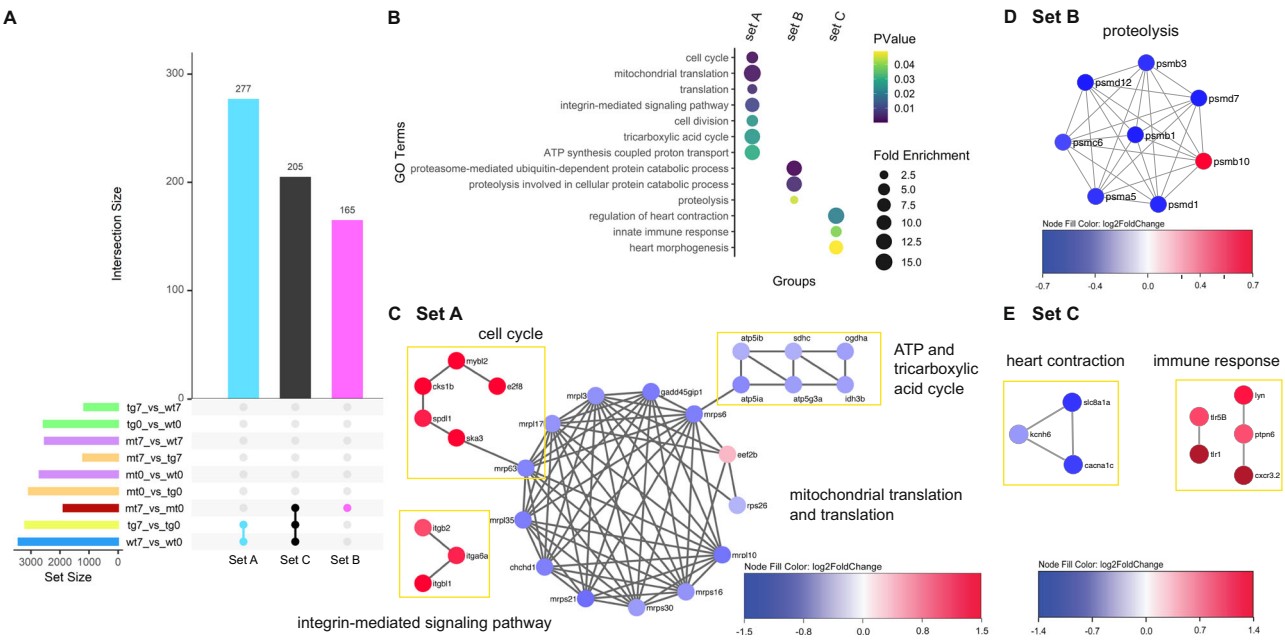

**Fig. 5 Network analysis underscores proliferative failure and proteolysis dampening in Nkx2.5 loss-of-function model. A** UpSet plot representing DEGs (FC > 0.5; FDR < 0.05) in nine relevant comparisons. Only selected interactions are illustrated: set A (turquoise) – only non-transgenic and transgenic wild-type-specific DEGs following injury, set B (magenta) – only *nkx2.5*$^{-/-}$-specific DEGs following injury, and set C (black) – DEGs common in non-transgenic wild-type, transgenic wild-type, and *nkx2.5*$^{-/-}$ fish following injury. **B** GO analysis of sets A, B, and C. All significantly enriched GO terms from each set are included for analysis by DAVID. Color and size of circles correspond to log (q-value) and enrichment, respectively. **C–E** Network analyses of genes associated with significantly enriched GO terms from set A (**C**), set B (**D**), and set C (**E**). Color represents the fold change in injured compared to uninjured wild-type CMs (WT7 vs WT0) (**C, E**) and injured compared to uninjured *nkx2.5*$^{-/-}$ CMs (MT7 vs MT0) (**D**). Red is upregulated and blue is downregulated, as shown in the key.

represent the injury response and are not confounded by differences in genotypes or controls (Fig. 5A). In this analysis strategy, the rows represent all appropriate two-way comparisons between the six individual samples (WT0, TG0, MT0, WT7, TG7, and MT7), summarizing the dataset size and delineating which comparisons are incorporated into the column sets. After contemplating appropriate inclusion and exclusion criteria, we narrowed in on three essential columns to scrutinize the reparative processes that are and are not initiated in the *nkx2.5*$^{-/-}$ fish independent of GRN aberrations from the loss of *nkx2.5* function. We identified set A (Fig. 5A, turquoise column) as those genes that are deployed in wild-type hearts following amputation but fail to be recruited in the absence of *nkx2.5* gene function. Set B (Fig. 5A, magenta column) denotes DEGs that are utilized in the *nkx2.5* loss-of-function model but are not activated in control scenarios. Finally, set C (Fig. 5A, black column) embodies the DEGs that are appropriately enlisted in the *nkx2.5*$^{-/-}$ fish, consistent with wild-type regenerative mechanisms.

The aforementioned set A encompasses 277 genes that fail to be activated in the *nkx2.5*$^{-/-}$ fish during cardiac regeneration (Fig. 5A, set A, turquoise column). This group is functionally enriched for several cellular processes including cell cycle, translation, integrin-mediated signaling pathway, and ATP synthesis coupled proton transport (Fig. 5B, set A). Further network analysis employing set A illuminates upregulation of the cell cycle and integrin-mediated signaling pathways, pinpointing critical, regenerative mechanisms that are boosted in the wild-type but not in the *nkx2.5*$^{-/-}$ fish following amputation (Figs. 5C and S7). Specifically, while wild-type fish mount a proliferative response and mediate extracellular matrix (ECM) production following amputation[19,40,42,43], *nkx2.5* loss-of-function model fails to mirror these cellular reactions. Moreover, downregulation of the tricarboxylic acid cycle and the ATP synthesis coupled

proton transport modules and similar dampening of the mitochondrial translation pathways in injured wild-type (but not *nkx2.5*-deficient) myocardium corroborate previous results emphasizing the necessary depression in metabolism during myocardial repair[13,44–46]. Further aligned with our findings that *nkx2.5*$^{-/-}$ fish demonstrate abnormal metabolism (Fig. 5C), recent murine studies describe an adult Nkx2-5 mutant model with significant abnormalities in mitochondrial respiratory pathways[47]. Altogether, analysis of set A accentuates that failure of the appropriate injury response in two essential cellular processes, CM proliferation and mitochondrial OXPHOS activity, is due to the loss of Nkx2.5 function.

Next, we queried the molecular pathways that are inappropriately regulated in the *nkx2.5*$^{-/-}$ fish and are not altered in the wild-type conditions (Fig. 5A, set B, magenta column). Set B encompasses 165 genes that are functionally enriched in proteasome-mediated ubiquitin-dependent protein and proteolysis involved in cellular protein catabolic processes (Fig. 5B, set B). Applying these GO terms, we produced a network depicting downregulation of several, closely linked members of the *psmd* gene family, proteosome 26S subunit, non-ATPase (Figs. 5D and S8). Proteasome-mediated ubiquitin-dependent protein catabolic processes degrade proteins in a precise and fastidious fashion employing the 19S regulatory subunit of the 26S proteasome[48]. More specifically, in the context of zebrafish cardiac regeneration, expression of *psmb1* and *psmb3* have been discovered in wound-edge CMs[49], evoking a functional role in sarcomere disassembly[12,49,50]. Thus, failure to stimulate expression of these proteolytic drivers may impair the ability of *nkx2.5*-deficient hearts to undergo dedifferentiation[12,13], an indispensable step in CM replenishment.

Finally, we investigated those genes which are differentially expressed in the same direction in non-transgenic wild-type,

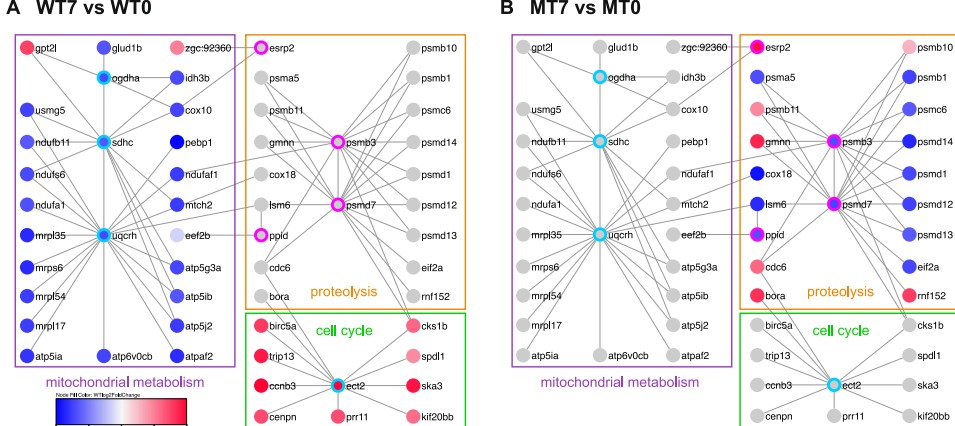

**Fig. 6 Network analysis of Nkx2.5 targets reveals distinct nodes that coordinate GRNs governing regenerative pathways. A, B** Network analyses of DamID targets in set A (turquoise border) and set B (magenta border). Color represents the fold change in injured compared to uninjured wild-type CMs (WT7 vs WT0) (**A**) and injured compared to uninjured $nkx2.5^{-/-}$ CMs (MT7 vs MT0) (**B**). Red is upregulated and blue is downregulated, as shown in the key. Mitochondrial metabolism modules are labeled in purple, proteolysis modules are labeled in orange, and cell cycles modules are labeled in green.

transgenic wild-type, and $nkx2.5^{-/-}$ fish in the uninjured and injured states at 7 dpa (Fig. 5A, set C, black column). Set C contains 205 genes appropriately engaged in the regenerative response in the $nkx2.5$ loss-of-function model. In this GO category, we found enrichment in the regulation of heart contraction and morphogenesis and in the innate immune response (Fig. 5B, set C; Figs. 5E and S9). These data emphasize that normal immunological pathways (Lai et al.[51]) are adequately stimulated in the $nkx2.5^{-/-}$ fish following amputation. Furthermore, while heart morphogenesis pathways downstream of Nkx2.5 are misregulated in the adult myocardium (Fig. 2D), these networks are not responsible for CM renewal in the injured heart. Taken together, our transcriptomic results indicate that reduction in regenerative potential in the $nkx2.5^{-/-}$ fish is due to the breakdown in three mechanistic injury responses vital for effective cardiac regeneration: cell cycle, mitochondrial metabolism, and proteolysis.

**Nkx2.5 targets identify nodes that link regenerative GRNs.** To determine the key effectors mediating these specific injury responses, we harnessed data from our previous studies to uncover a consolidated set of Nkx2-5 direct targets identified in mouse HL-1 cells by DamID[52]. We then intersected our RNA-seq results with these DamID targets to identify the key GRNs nodes that function immediately downstream of Nkx2-5 to promote regeneration. To perform this overlap, we ascertained the zebrafish orthologues of the EnsEMBL IDs of the mouse DamID targets using BioMart (release 100)[53]. Overlapping these datasets with the entire gene lists in sets A and B from our RNA-seq results (Fig. 5A; independent of GO annotation) highlights interconnecting nodes that are under the control of Nkx2-5 and link the enriched modules for cell cycle, mitochondrial metabolism, and proteolysis (Fig. 6). When comparing uninjured and injured non-transgenic wild-type hearts (Fig. 6A), our analysis supports prior data by exhibiting appropriately diminished ATP-dependent and translational pathways and stable protein catabolic processes to support CM renewal[46,54]. Moreover, proliferative genes are triggered following injury to simulate repair (Fig. 6A)[17]. Importantly, Nkx2.5 effectors coordinate these GRNs through critical hubs, such as *ect2*, *psmb3*, and *psmd7*, which assemble a consolidated response to injury. Alternatively, when evaluating the uninjured and injured $nkx2.5^{-/-}$ fish (Fig. 6B), these crucial modules either fail to be inhibited, in the case of mitochondrial translation, or fail to be activated, in the case of proliferation. Further, inappropriate downregulation of the genes

required for sarcomere disassembly exacerbates the collapse in reparative mechanisms (Fig. 6B). Altogether, integration of our new RNA-seq dataset and our existing DamID target list uncovers strategic nodes by which Nkx2.5 synchronizes the adult myocardial injury response.

To inspect the biological significance of these GRNs, we implemented in vivo analyses to detect expression of crucial nodes in non-transgenic wild-type and $nkx2.5^{-/-}$ fish at 7 dpa. *ect2*, a putative direct target of Nkx2.5 from our DamID study and confirmed to be bound by NKX2-5 in human pluripotent cell-derived cardiomyocytes[55], is well positioned to promote cell cycle re-entry and sarcomere disassembly during cardiac regeneration given the communication that it establishes between these two modules (Fig. 6A, B). Previous studies have elucidated the function of *ect2* in contractile ring assembly and initiation of cytokinesis; Ect2 activity converts RhoA-GDP (guanosine diphosphate) to RhoA-GTP (guanosine triphosphate) at the spindle[56]. Employing transient dominant negative expression in zebrafish, recent reports have illuminated the role of *ect2* loss-of-function in inhibiting cytokinesis, inducing polyploidization, and ultimately, in yielding a loss of cardiac regenerative potential[17]. Applying RNAscope technology, we documented upregulation of *ect2* expression in the injury zone and wound edge of injured, non-transgenic wild-type apices (Fig. 7A), validating the importance of Ect2 function in CM cytokinesis. Moreover, *ect2* expression is significantly depleted in injured $nkx2.5^{-/-}$ hearts which supports our hypothesis that Nkx2.5 is required upstream of *ect2* in promoting cell cycle reentry and proliferation in regenerating myocardium (Fig. 7B). Finally, our conclusion is further reinforced by evidence that $nkx2.5$ and *ect2* transcripts co-localize in individual nuclei at the injury border (Fig. 7C, D), suggesting cell autonomous interactions.

Two additional Nkx2.5 effectors, *psmb3* and *psmd7*, sit high in the Nkx2.5 transcriptional hierarchy linking the mitochondrial translation and the proteolysis modules (Fig. 6A, B). Given data depicting *psmb3* and *psmb1* transcripts in the wound edge following cardiac injury in wild-type zebrafish[49], downregulation of these proteasome gene family members has the potential to prevent proteolytic mechanisms necessary to disassemble mature CM architecture. We postulate that Nkx2.5 regulates *psmb3* and *psmd7* expression following ventricular injury and, in the $nkx2.5$ loss-of-function model, depleted *psmb3* and *psmd7* expression restrains dedifferentiation of pre-existing CMs. Employing ISH, we document robust expression of *psmb3* (Fig. 7E) and *psmd7*

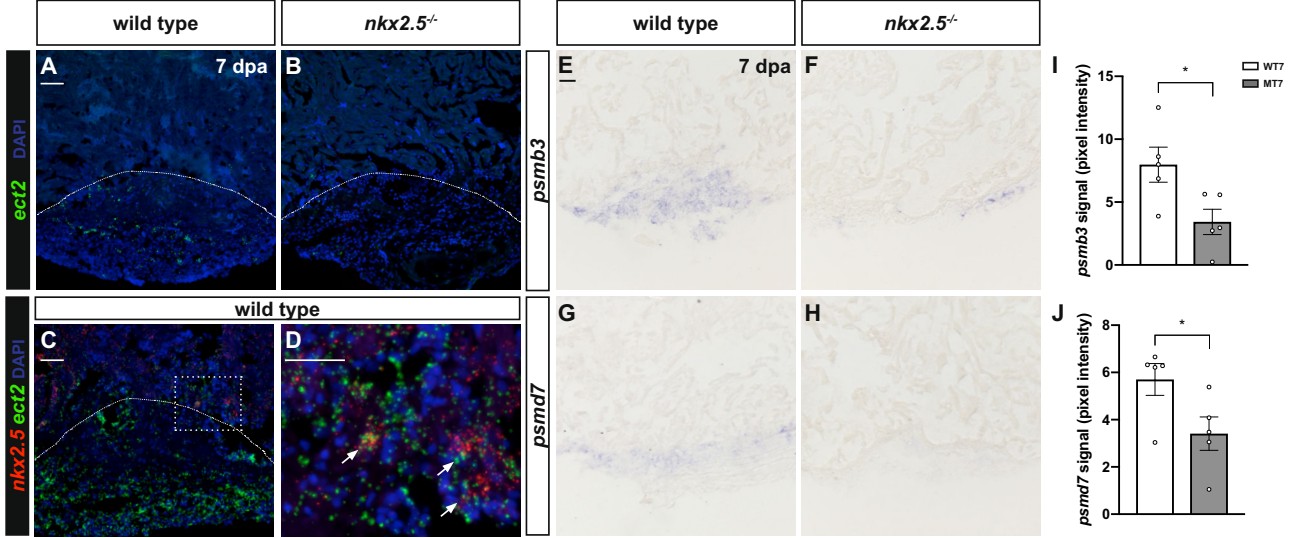

**Fig. 7 Nkx2.5 regulates targets, *ect2*, *psmb3*, and *psmd7*, to orchestrate regenerative mechanisms. A**, **B** Representative images of RNAscope analysis on sections of injured hearts at 7 dpa display decreased *ect2* expression in the regenerating myocardium of the *nkx2.5$^{-/-}$* ($n = 7$) (**B**) compared to the non-transgenic wild-type ($n = 3$) (**A**) fish. Dashed lines represent amputation planes. Scale bar, 50 μm. **C**, **D** Additional RNAscope analysis employing both *nkx2.5* and *ect2* probes simultaneously highlight co-localization of transcripts in individual nuclei at the injury border zone in non-transgenic wild-type fish ($n = 3$) (**C**). Higher magnification panel (**D**) indicates the area outlined in the rectangular box. Scale bar, 50 μm. **E–H** In situ hybridization for *psmb3* (**E**, **F**) and *psmd7* (**G**, **H**) in non-transgenic wild-type ($n = 5$) (**E**, **G**) and *nkx2.5$^{-/-}$* ($n = 5$) (**F**, **H**) adult hearts. In *nkx2.5$^{-/-}$* hearts, both *psmb3* and *psmd7* are downregulated in the injured area. Scale bar, 30 μm. **I**, **J** Quantification of *psmb3* (**I**) and *psmd7* (**J**) signal, detected by in situ hybridization, reveals statistically significant diminution of both genes in the *nkx2.5$^{-/-}$* ($n = 5$) compared with the wild-type ($n = 5$) injured myocardium. Mean and standard error of each data set are shown. Unpaired, two-tailed *t*-test illustrate statistically significant differences in (**I**) ($p = 0.0297$) and (**J**) ($p = 0.0468$). Source data are provided as a Source Data file.

(Fig. 7G) in the regenerate of non-transgenic wild-type hearts. Strikingly, there is diminished expression of these proteasome genes in the wound of *nkx2.5$^{-/-}$* fish (Fig. 7F, H). Moreover, pixel intensity quantification corroborates these histological findings; both *psmb3* and *psmd7* exhibit a statistically significant decrease in expression in the wound (Fig. 7I, J). Altogether, we conclude that Nkx2.5 operates via key effectors, *ect2*, *psmb3*, and *psmd7*, to orchestrate regenerative mechanisms involving mitochondrial metabolism, dedifferentiation, and cytokinesis to repopulate CMs following injury.

**Nkx2.5 promotes proliferation and dedifferentiation.** Considering this defective transcriptional response in *nkx2.5$^{-/-}$* fish, we examined the source of the impaired healing in vivo by evaluating the CM proliferation indices at 7 dpa and 14 dpa in wild-type and *nkx2.5$^{-/-}$* fish. Given CM-specific *nkx2.5* expression (Fig. 3), we hypothesized that Nkx2.5 is required cell autonomously to induce dedifferentiation, cell division, and redifferentiation of new CMs. Our data reveal a statistically significant decrease in the CM proliferation index in animals with reduced *nkx2.5* transcriptional activity versus animals with intact *nkx2.5* function at 7 dpa (Fig. 8A, B, E). In contrast, there is no statistically significant discrepancy at 14 dpa in the proliferative rates between wild-type and *nkx2.5$^{-/-}$* fish (Fig. 8C–E). These findings illuminate an essential role for Nkx2.5 in promoting CM proliferation at 7 dpa when an escalation in cellular renewal is required for myocardial regeneration, validating the GRNs created from the RNA-seq data (Figs. 5C and 6).

We next examined the injured *nkx2.5$^{-/-}$* fish for evidence of impaired sarcomere disassembly given downregulation of the proteolysis module of the Nkx2.5-dependent GRN (Figs. 5D and 6). Prior work has detected reactivation of embryonic cardiac sarcomeric proteins at the wound edge, defining embryonic cardiac myosin heavy chain (embCMHC) as a marker of

undifferentiated CMs[57]. To assess whether *nkx2.5* regulates dedifferentiation and, therefore, re-expression of immature myosins, we performed immunostaining with embCMHC and MF20 at 7 dpa (Fig. 8F, G). While wild-type hearts demonstrate co-localization of embCMHC and MF20 along the border of the regenerate (Fig. 8F), *nkx2.5$^{-/-}$* myocardium shows minimal expression of embCMHC at the site of injury (Fig. 8G). These findings are further corroborated with immunostaining directed against activated leukocyte adhesion molecular a (Alcam), another marker of dedifferentiation[58,59], demonstrating diminished Alcam expression in the *nkx2.5$^{-/-}$* compared to the wild-type regenerates (Fig. S10). From these data, we conclude that adult CMs fail to dedifferentiate in the absence of *nkx2.5* gene function. Taken together, our results validate GRNs underscoring that Nkx2.5 is required cell autonomously to instruct proteolytic pathways for dedifferentiation of mature CMs to an immature myocardial fate (Figs. 5D and 6).

**Nkx2.5 non-cell autonomously limits epicardial integration.** Although our data establish that CM-specific *nkx2.5* expression directly regulates CM proliferation and dedifferentiation via Nkx2.5-dependent cell cycle and proteolysis modules, we sought to evaluate whether Nkx2.5 also mediates non-cell autonomous regenerative functions. Both the cardiac endocardial and epicardial lineages exhibit sentinel roles in heart regeneration and interaction among these specific cell types allows for orchestration of the complex healing process. Coronaries arteries are required to regenerate in order to perfuse new myocardium[23] and revascularization has been shown to occur within hours of cryoinjury to stimulate CM proliferation[41]. Thus, using *nkx2.5$^{-/-}$* fish carrying *Tg(kdrl:EGFP)$^{la116}$*, we investigated the process of angiogenesis in wild-type and *nkx2.5* loss-of-function animals. Interestingly, we observed normal EGFP$^+$ endothelial tubes in the wound area

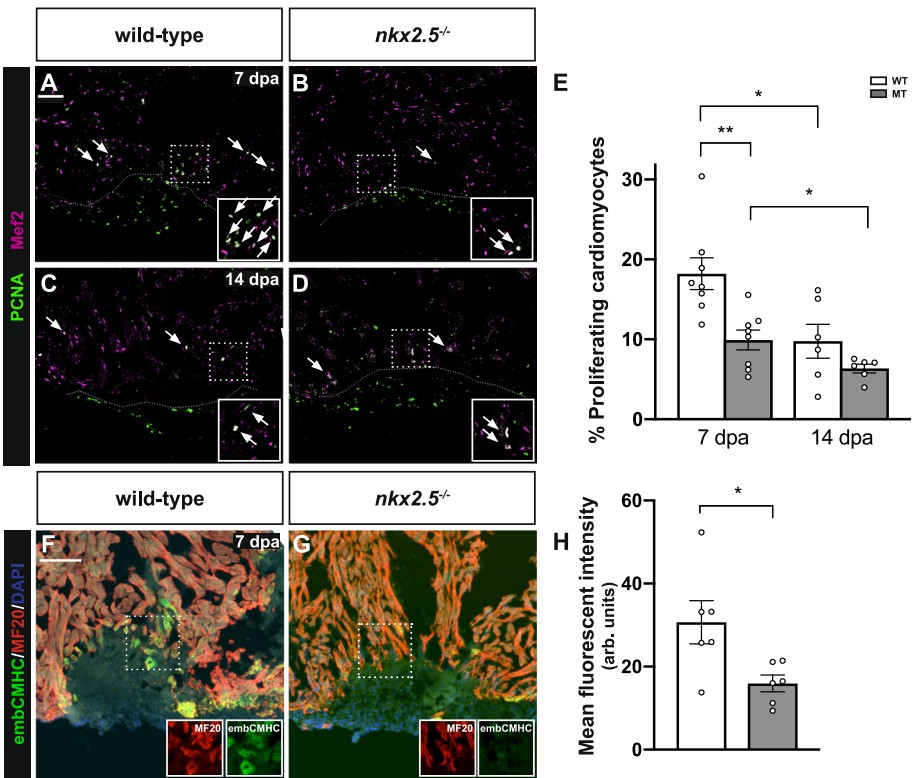

**Fig. 8 Nkx2.5 is required for cardiomyocyte proliferation and dedifferentiation in the regenerating heart. A–D** Representative images of PCNA/Mef2 immunostaining in wild-type ($n = 8$ in **A**, $n = 6$ in **C**) and $nkx2.5^{-/-}$ ($n = 8$ in **B**, $n = 6$ in **D**) hearts indicate proliferating CMs (arrows). Scale bar, 100 µm. **E** Quantification of PCNA⁺Mef2⁺ cells in wild-type and $nkx2.5^{-/-}$ hearts. $n = 8$ at 7 dpa and $n = 6$ at 14 dpa for both genotypes. Mean and standard error of each data set is shown. Unpaired, two-tailed $t$-tests show statistical differences between WT7 and MT7 ($p = 0.0033$), WT7 and WT14 ($p = 0.0143$), and MT7 and MT14 ($p = 0.0369$). **F, G** Representative images of embCMHC/MF20 immunostaining illustrating diminished expression of immature myosin in the $nkx2.5^{-/-}$ ($n = 6$) compared to the wild-type ($n = 6$) wound border. Scale bar, 100 µm. **H** Quantification of integrated signal density of the embCMHC stain is depicted in wild-type ($n = 6$) and $nkx2.5^{-/-}$ ($n = 6$) injury borders. Mean and standard error of each data set are shown. Unpaired, two-tailed $t$-test indicates a statistically significant difference ($p = 0.0248$). Source data are provided as a Source Data file.

establishing that $nkx2.5$ expression is not responsible for coronary angiogenesis during cardiac regeneration (Fig. S11).

Next, we compared the regenerative response in the epicardium, a thin mesothelial cell layer overlying the atrial and ventricular chambers, between wild-type and $nkx2.5^{-/-}$ fish. Previous studies have elucidated the essential roles of epicardial activation, proliferation, and colonization of the regenerate[23]. Specifically, epicardial cells undergo an epithelial-to-mesenchymal (EMT) transition to yield epicardial derived cells (EPDCs). These EPDCs invade the injury site between 7 dpa and 30 dpa to provide a cellular scaffold and produce perivascular cells and myofibroblasts, ultimately in support of CM proliferation and angiogenesis[39,60]. As the wound heals, the epicardium undergoes EMT and establishes a thickened epicardial layer covering the injured myocardium[60–64]. Employing immunostaining with Tg(tcf21:DsRed2) to label the epicardium in wild-type and $nkx2.5^{-/-}$ fish, we discerned expression at the injury site in wild-type and $nkx2.5^{-/-}$ fish at 14 dpa (Fig. 9A–D). Yet, impaired integration of $tcf21^+$ cells in the $nkx2.5^{-/-}$ compared to wild-type fish is evident; $tcf21^+$ epicardial cells fail to penetrate the regenerate and remain exterior to the CM layer (Fig. 9B, D). Quantification of EPDC migration reveals a statistically significant decrement in the distance measured from the apex of the heart to the maximal interior position of the Tcf21:DsRed2⁺ cells when comparing wild-type and $nkx2.5^{-/-}$ hearts (Fig. 9E). Taken together, these data indicate that, while the epicardium mounts a suitable response to tissue damage in the loss-of-function $nkx2.5$

model, compromised epicardial penetration into the wound weakens the healing potential in the $nkx2.5^{-/-}$ compared to wild-type fish.

We sought to clarify the underlying cellular mechanism responsible for this deficient epicardial infiltration into the regenerate. Specifically, we tested our hypothesis that Nkx2.5 mediates non-cell autonomous functions in addition to the CM-intrinsic roles of dedifferentiation and proliferation. Thus, we performed ex vivo assays using non-transgenic wild-type and $nkx2.5^{-/-}$ fish carrying $Tg(tcf21:dsRed)$, as previously described[65]. At 3 dpa, we resected the ventricular apices, cultured them in fibrin-coated plates, and quantified the outgrowth of epicardial cells after four days. There were no significance differences identified following quantification of epicardial cell expansion between the wild-type and the $nkx2.5^{-/-}$ cultured apices (Fig. 9F–H). Moreover, employing EdU incorporation, we established that the proliferation rate in the wild-type epicardial cultured cells is the same as that in the $nkx2.5^{-/-}$ epicardium (Fig. 9I–K). These findings support our conclusion that the defect in epicardial infiltration of the regenerate is due to mechanisms extrinsic to the epicardial cells. This conclusion aligns elegantly with our network analyses which invoke a small, integrin-mediated signaling pathway module ($itgb2$, $itba6a$, and $itgbl1$) that fails to be stimulated in the $nkx2.5$ loss-of-function model (Fig. 5C). Given the importance of ECM remodeling in cardiac regeneration[43,45,66], we expect that future investigation of targets of Nkx2.5 will elucidate novel ECM regulatory networks

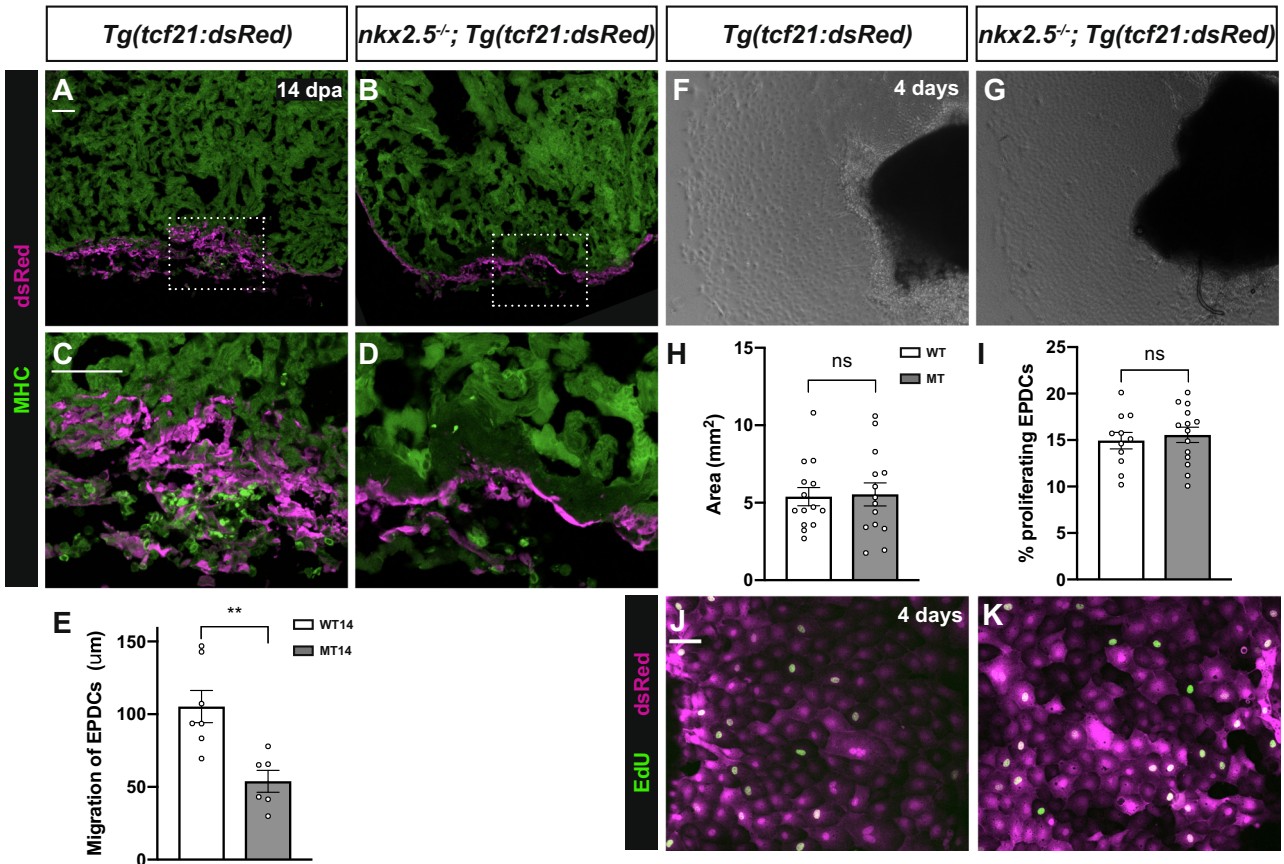

**Fig. 9 Impaired epicardial migration in the *nkx2.5* loss-of-function model. A–D** Myosin Heavy Chain (MHC) immunostaining in injured *Tg(tcf21:dsRed)* (*n* = 6) (**A, C**) and *nkx2.5⁻/⁻;Tg(hsp70l:nkx2.5-EGFP);Tg(tcf21:dsRed)* (*n* = 5) (**B, D**) at 14 dpa exhibits aborted epicardial penetration in the *nkx2.5⁻/⁻* heart. Scale bar, 50 μm. **E** EPDC migration quantification strategy measures the distance from the apex of the heart to the maximal interior position of the Tcf21:DsRed2⁺ cells in injured wild-type (*n* = 6) and *nkx2.5⁻/⁻* (*n* = 5) hearts (**A–D**). Mean and standard error of each data set are shown. Unpaired, two-tailed *t*-test demonstrates a statistically significant difference between WT14 and MT14 (*p* = 0.0035). **F, G** Ex vivo epicardial migration assay in *Tg(tcf21:dsRed)* (*n* = 14) (**G**) and *nkx2.5⁻/⁻;Tg(hsp70l:nkx2.5-EGFP);Tg(tcf21:dsRed)* (*n* = 14) (**H**) explants. **H, I** Quantification of the epicardial migration showing no statistical difference between the wild-type (*n* = 14) and the *nkx2.5⁻/⁻* (*n* = 14) cultured apices (**F, G**). Additional quantification of Edu⁺tcf21:dsRed⁺ cells indicates normal proliferative potential in cultured *nkx2.5⁻/⁻* (*n* = 14) versus wild-type epicardium (*n* = 12) (**J, K**). Mean and standard error of each data set are shown with no statistically significant differences detected by unpaired, two-tailed *t*-tests between WT and MT (*p* = 0.8794 in **H** and *p* = 0.6165 in **I**). **J, K** Edu proliferation assay in wild-type (*n* = 12) (**K**) and *nkx2.5⁻/⁻* (*n* = 14) (**L**) epicardial tcf21:dsRed⁺ cells. Scale bar, 50 μm. Source data are provided as a Source Data file.

exemplifying non-cell autonomous functions of this cardiac-specific transcription factor.

## Discussion

These data offer a unique perspective on mechanisms regulated by Nkx2.5 that are responsible for building new muscle following injury in the mature zebrafish heart. In particular, we show that CM-specific expression of *nkx2.5* is required for cardiac regeneration following amputation of the ventricular apex in an innovative, adult Nkx2.5 loss-of-function zebrafish model. Using this genetic tool, we harnessed both new RNA-seq and our prior DamID data to reveal that the developmental molecular profile regulated by Nkx2.5 is perturbed in the adult *nkx2.5*-deficient myocardium. Moreover, our multi-variate network analysis underlines Nkx2.5 as a vital regulator of CM renewal, functioning to promote cell cycle re-entry and to maintain proteolysis genetic modules. Direct targets, *ect2*, *psmb3*, and *psmd7* provide examples of essential hubs in the Nkx2.5-dependent GRNs guiding proliferation, sarcomere disassembly[67,68], and mitochondrial metabolism[47]. We observe cellular evidence of these molecular mechanisms in the diminished number of PCNA⁺ CMs and the decreased expression of immature cardiac myosin and Alcam in

the injury border of *nkx2.5⁻/⁻* hearts. Yet, additional non-cell autonomous functions of Nkx2.5 are necessary for epicardial penetration into the wound to support the reparative micro-environment niche. Moreover, ECM modules fail to be upregulated in the absence of *nkx2.5* gene function, suggesting a paradigm whereby Nkx2.5 mediates crucial components in ECM patterning. Taken together, these studies underscore the importance of activation of embryonic Nkx2.5 for CM dedifferentiation and proliferation during cardiac regeneration and suggest opportunities for therapeutic intervention in patients suffering from myocardial infarction and adult CHD.

While it is clear that evolutionarily conserved developmental genetic signatures are critical to facilitate regenerative responses from cuttlefish to mouse[69–73], the toolkits required to investigate the underlying essential mechanisms remain underdeveloped. There is a scarcity of genetic lines with *loxP* sites flanking genes of interest that limits our ability to institute conditional deletions efficiently. Furthermore, while some pivotal cardiac transcription factors have been examined functionally[21,22,74], many members of the cardiac developmental transcriptional hierarchy have yet to be investigated. Although murine studies have benefitted from conditional deletion of *Nkx2-5* to examine its roles in the adult

cardiac conduction system and trabecular myocardium[75,76], the *Nkx2-5* requirement in myocardial repair was not addressed due to the limited regenerative potential in these models. Here, we describe the application of an innovative technique expediting rescue of otherwise lethal *nkx2.5⁻/⁻* embryos using heat shock overexpression without confounding evidence of non-specific defects[29]. We have taken advantage of our *nkx2.5*-deficient fish to dissect the cell type-specific functional requisites, uncovering inappropriate responses in proliferation, sarcomere disassembly, and mitochondrial metabolic pathways in the CMs and cell extrinsic abnormalities in ECM deposition of the regenerative niche. This unique model provides a substrate to study the distinct molecular profiles that are altered in the Nkx2-5 loss-of-function myocardium, mimicking late-stage challenges faced by adult patients with Nkx2-5-associated CHD.

Our data also illustrate insights generated from a comprehensive assessment of the injured Nkx2.5 loss-of-function adult model and multi-variate network interpretation of transcriptional profile alterations in various genetic and temporal conditions. By probing each comparison, we identified distinct responses – both appropriate and inappropriate – that are regulated by Nkx2.5. These findings suggest that therapeutic measures need to be tailored to the individual clinical scenarios in order to ameliorate pathological mechanisms and redirect the interlocked chain of events that ensue following myocardial damage[73]. Specifically, in the context of adult CHD patients carrying *NKX2-5* mutations, we envisage that discrete and directed remedies, such as stimulation of cell cycle re-entry and suppression of mitochondrial translation, would augment successful regenerative outcomes. Moreover, the proteasome-mediated ubiquitin-dependent protein catabolic processes necessitate preservation to ensure proper dedifferentiation of Nkx2.5-deficient CMs. Our data suggest that modulation of the rapid inflammatory response following injury occurs aptly and, therefore, does not need remediation to support effective repair. Taken together, careful delineation of the molecular and cellular processes that go awry following injury in both wild-type and genetically altered myocardium will enhance our understanding of regenerative strategies and permit recruitment of explicit endogenous mechanisms deficient in each patient-specific context.

While evidence of CM-specific expression of *nkx2.5* in the adult zebrafish heart, diminished proliferation, and arrested dedifferentiation illuminate cell intrinsic functions of Nkx2.5 following injury, impaired epicardial migration in vivo points to the contribution of CM extrinsic roles to the decreased reparative capability in *nkx2.5⁻/⁻* fish. In light of our ex vivo studies depicting normal epicardial migration and proliferation and our network analyses illustrating failure of *nkx2.5⁻/⁻* fish to upregulate an integrin-mediated ECM module, we speculate that Nkx2.5 directly or indirectly regulates crucial constituents of the cardiac matrix. Further investigation is required to uncover the key players acting downstream of Nkx factors. Yet, recent reports indicate that distinct embryonic ECM cargo can augment cytokinesis in postnatal rat CMs in vitro and in vivo[77], providing a glimpse of therapeutic potential. Moreover, zebrafish studies have advanced our appreciation of the mechanisms responsible for the cardiac valve regeneration[78,79], further emphasizing the importance of studying ECM regulation in adult Nkx2.5 loss-of-function models given the incidence of valvular defects in these patients[80,81]. To address questions on the impact of Nkx2.5-regulated ECM on cardiac and valvular repair, future inquires of the regenerative proteome with emerging techniques is necessary to understand global protein-protein interaction dynamics[82,83].

Altogether, our studies underscore that activation of the embryonic Nkx2.5 transcriptional network is critical in triggering productive intrinsic and extrinsic cellular responses to myocardial damage in the adult zebrafish heart. Exploiting insights into the specific mechanistic processes that are hampered in the *nkx2.5⁻/⁻* fish such as CM proliferation and dedifferentiation, our work open doors for the cultivation of targeted therapies in Nkx2-5-deficient myocardium. Studies demonstrating opportunities for myriad factors to instruct CM regeneration indicate that therapeutic intervention is feasible[84–86] as long as the coordination of systematic approaches is realized[73]. The identification of previously unrecognized Nkx2.5 targets, *ect2*, *psmb3*, and *psmd7*, as critical nodes in orchestrating cell cycle re-entry, proteolysis, and mitochondrial metabolism in response to injured myocardium offers initial steps to achieve this mission.

## Methods

**Zebrafish lines and ventricular resections**. We used adult zebrafish carrying the following previously described mutation and transgenes: *nkx2.5ᵛᵘ¹⁷⁹*[31], *Tg(hsp70l:nkx2.5-EGFP)ᶠᶜᵘ¹²⁹*, *Tg(nkx2.5:ZsYellow)ᶠᵇ⁷⁸⁷*, *Tg(tcf21:DsRed2)ᵖᵈ³⁷³⁹*, and *Tg(kdrl:EGFP)ˡᵃ¹¹⁶⁸⁸*. Adult *nkx2.5⁻/⁻;Tg(hsp70l:nkx2.5-EGFP)* fish were generated as previously described[29] and experiments were implemented with one transgenic parent per cross. Ventricular apices were resected in adult fish (3- to 18-months-old)[17], employing a previously described technique[38]. All the zebrafish experiments were performed according to the protocol approved by the Institutional Animal Care and Use Committee (IACUC) at Columbia University.

**Heat shock conditions**. Embryos from outcrosses of fish carrying *Tg(hsp70l:nkx2.5-EGFP)* were maintained at 28.5 °C and exposed to heat shock at 37 °C at 21 somites. To implement heat shock, 50 embryos were placed in 2.5 mL of preheated embryo medium in a 35 mm Petri dish on top of a covered heat block for one hour. Three hours following initiation of heat shock, transgenic embryos were identified by visualization of ubiquitous EGFP expression. Non-transgenic sibling embryos exposed to heat shock served as controls.

**Histological analysis**. Zebrafish were anesthetized with tricaine and then fixed in 4% paraformaldehyde overnight at 4 °C. Subsequently, the dissected hearts were washed several times with PBS and cryopreserved with 30% sucrose before immersion in O.C.T. (Tissue-Tek). The blocks were stored immediately at −80 °C. They were sectioned at 10 μm with a cryostat (Leica CM3050 S) and collected for Acid Fuchsin Orange G (AFOG), RNA in situ hybridization (ISH), RNAscope, and immunostaining.

AFOG staining was performed as previously described on 6–10 animals per time point[38]. Briefly, slides were dried for one hour at room temperature and re-fixed with 4% PFA for five minutes. Subsequently, they were incubated in a preheated Bouin's solution for two hours at 60 °C followed by one hour at room temperature. After extensive washes with H₂O, the slides were rinsed in 1% phosphomolybdic acid for five minutes. Following a five-minute wash with H₂O, they were incubated for 10 min in AFOG staining solution (0.5% Aniline Blue, 1% Orange G, 1.5% Acid Fuchsin, pH 1.09). Finally, the slides were rinsed in H₂O, dehydrated, cleared in xylene, and mounted in Cytoseal 60 (Thermo Fisher Scientific).

ISH was performed as previously described with minor modifications[89]. Briefly, slides were brought to room temperature for one hour before performing post-fixation in 4% PFA for 20 min. Subsequently, they were washed with PBS, treated with proteinase K (10 μg/ml) for 10 min at 37 °C, washed again with PBS prior to another 4% PFA fixation for 5 min, washed again with PBS, and treated with HCl 0.07 N for 15 min of shaking. Then, the sections were washed with PBS and a pre-hybridization buffer (0.2 M NaCl, 5 mM EDTA, 12 mM Tris HCl pH7.5, 14 mM Na₂HPO₄-7H₂0, 6 mM NaH₂PO₄, 50% formamide, 10% dextran sulfate, 10 mg/ml yeast tRNA, 1x Denhardt's) at 65 °C for two hours. Hybridization of the probe was performed overnight at 65 °C, covered with a coverslip in a humidified chamber. The following day, the sections were washed three times with solution I (50% formamide, 5x SSC, 0.1% Tween-20) for 30 min at 65 °C and three more times with solution II (50% formamide, 2x SSC, 0.1% Tween-20) for another 30 min, each at 65 °C. Subsequently, the sections were washed three times with maleic acid buffer (MAB) with 0.1% Tween-20 (MABT) for ten minutes at room temperature. The sections were then incubated in a blocking solution (70% MABT, 20% Normal Goat Serum, and 10% blocking reagent (Roche)) for two hours at room temperature. Next, the anti-DIG-AP antibody (11093274910, Roche; 1:2000) was applied in blocking solution overnight at 4 °C with a coverslip in a humidified chamber. On the following day, the slides were washed with MABT for ten minutes, again for three one-hour-long washes, and finally three washes with alkaline-phosphate buffer (100 mM NaCl, 50 mM MgCl₂, 100 mM Tris pH 9.5, 0.1% Tween-20). The sections were developed in BM-purple (Roche) at 37 °C until signal was detected. The following probes were used: *amhc* (*myh6*; ZDB-GENE-031112-1)[90], *vmhc* (*myh7*; ZDB-GENE-991123-5)[91], *psmb3* (ZDB-GENE-040426-2682), and *psmd7* (ZDB-GENE-030131-5541).

Immunostaining was performed as previously described[92]. Briefly, the hearts were collected in cold PBS and brought to room temperature. Following washes

with PBS plus 0.1% Tween20 (PBT), sections were permeabilized with methanol for one hour at room temperature, blocked in blocking solution (PBS containing 2% sheep serum, 0.2% Triton X-100, 1% DMSO) for 30 min at room temperature, and incubated with primary antibodies in blocking solution at 4 °C overnight. Sections were then incubated with secondary antibodies between one and three hours at room temperature. Finally, the slides were mounted with Vectashield medium (Vector Laboratories) and DAPI prior to imaging.

Mef2/PCNA staining was performed as previously described[93]. Briefly, the slides were equilibrated to room temperature prior to performing antigen retrieval in 10 mM citrate buffer at 98 °C for 20 min. Then, they were cooled for 20 min, rinsed in $H_2O$ twice, and immersed in pre-cooled acetone (−20 °C) for 10 min. Subsequently, acetone was allowed to evaporate for 20 min at room temperature. The slides were then washed once in $H_2O$, washed twice in PBA, and incubated in blocking buffer (5% normal goat serum, 0.3% Triton X-100 in PBS) for one hour at room temperature. Incubation with primary antibody was performed in antibody dilution buffer (1% BSA, 0.3% Triton X-100 in PBS) overnight at 4 °C, and secondary antibody for two hours at room temperature.

Primary antibodies used were: anti-PCNA (WH0005111M2, Sigma, 1:200), anti-Mef2 (sc-313, Santa Cruz Biotechnology 1:50), anti-tropomyosin (CH1, Developmental Studies Hybridoma Bank, 1:100), anti-myosin heavy chain (F59, Developmental Studies Hybridoma Bank, 1:25), anti-GFP (A-11122, Invitrogen, 1:200), anti-DsRed (632496, Clontech, 1:200), anti-raldh2 (GTX124302, GeneTex, 1:500), anti-vimentin (40E-C, Developmental Studies Hybridoma Bank, 1:35), anti-embCMHC (N2.261, Developmental Studies Hybridoma Bank, 1:50), anti-myosin heavy chain (MF20, Developmental Studies Hybridoma Bank, 1:20), and anti-Alcam (ZN-8, Developmental Studies Hybridoma Bank, 1:10). The following secondary antibodies were used (1:500): Alexa Fluor 488 Goat anti-Mouse IgG1 (A-21121, Invitrogen), Alexa Fluor 488 Goat anti-Mouse IgG2a (A-21131, Invitrogen), Alexa Fluor 568 Goat anti-Mouse IgG1 (A-21124, Invitrogen), Alexa Fluor 568 Goat anti-Mouse IgG2b (A-21144, Invitrogen), Alexa Fluor 568 Goat anti-Rabbit IgG (A-11011, Invitrogen), Alexa Fluor 568 Goat anti-Mouse IgM (A-21043, Invitrogen), and Alexa Fluor 633 Goat anti-Rabbit IgG (A-21070, Invitrogen).

RNAscope staining was executed following the manufacturer's instructions for the RNAscope multiplex fluorescent assay v2 (Advanced Cell Diagnostics or ACD). RNAscope probes for *vcana* (C3), *nkx2.5* (C2), and *ect2* (C1) were synthesized by ACD and nuclei were stained with DAPI prior to analysis.

Images for AFOG, ISH, and RNAscope were acquired with a Zeiss Axio Imager.D2m and a Hamamatsu Nanozoomer SQ. Following immunostaining protocols, confocal imaging was performed with a Nikon Ti Eclipse inverted confocal microscope and a Zeiss LSM 880 with z-stacks analyzed using ImageJ (version 1.53k, NIH).

**Quantification and statistical analysis.** All quantifications were performed blinded. Unless otherwise indicated, experiments were implemented with at least four biological replicates and at least three samples per replicate. For quantification strategies, three non-consecutive sections with the largest area were measured to assign an index for each heart. To calculate the proportion of scar tissue, the section with the largest injury site was selected and normalized to the ventricular area. Furthermore, the severity of the scar was classified as previously described[94]. Ventricular volume was calculated on 10 μm sections by totaling the ventricular area in each section and multiplying by the distance between each section[32]. The percentage of trabeculation was calculated as the percentage of muscular tissue within the ventricular area using the threshold tool and manually excluding the internal cavity area[32]. For quantification of the ISH signals, previously reported protocols were executed[95]. For fluorescent intensity calculations, embCMHC+ and Alcam+ areas were quantified in pixels employing ImageJ in the region of the regenerate and normalized to background. To measure proliferation, Mef2+/PCNA+ cells in a 200 μm area from the border were counted manually[96]. Individual CMs were categorized by the presence of Mef2 and by a size greater than 5 μm diameter. Of these CMs, those cells also positive for PCNA signal were counted and averaged to calculate a proliferative index for each heart. In vivo migration of epicardial derived cells (EPDCs) was quantified by measuring the distance between the ventricular apex and the furthest EPDC inside the myocardium. For statistical analyses, two-tailed Student's *t*-tests were used when normality test was passed. Statistical values are displayed as mean ± standard error of the mean (SEM). The following nomenclature was employed to present results: ns not significant, *$p < 0.05$, **$p < 0.01$; ***$p < 0.001$, ****$p < 0.0001$.

**Quantitative PCR and RNA sequencing.** For qPCR, total RNA was extracted from six pooled ventricles for each condition and timepoint analyzed. The purified RNAs were reverse transcribed with the iScript™ cDNA Synthesis Kit (Bio-Rad). qPCR was performed with iQ™ SYBR Green Supermix (Bio-Rad) and a CFX96 Touch™ Real-Time PCR Detection System (Bio-Rad). All experiments were completed using biological triplicates. Primer sequences are noted in Supplemental Table 1.

For RNA Sequencing, total RNA from six pooled ventricular apices of two biological replicates was extracted using TRIzol reagent (Invitrogen) according to the manufacturer's instructions. Next, the quality was assessed using a Bioanalyzer. The cDNA library was prepared using TruSeq RNA Prep Kit v2 (Illumina) and the samples were submitted to the JP Sulzberger Columbia Genome Center at

Columbia University Medical Center. A HiSeq 2500 System was used to obtain single-end 100 bp reads for each sample.

**RNA-sequencing data processing and analysis.** Reads were aligned to the GRCz10-91 zebrafish reference genome using STAR (version 2.5.2)[97] and were quantified using featureCounts (version 1.5.0-p3)[98]. Differential gene expression between conditions was performed using DEseq2 (version 3.14) (Anders and Huber, 2012). Statistically significant differentially expressed genes (DEGs) were retrieved by filtering for genes with absolute log2 fold-change (FC) > 0.5 and false discovery rate (FDR) < 0.5 in R (version 3.6.0).

Venn overlaps of DEGs were generated with the VennDiagram package (version 1.7.1)[99]. Hierarchical clustering was performed using the gplots::heatmap.2() function which employs Euclidean distance measures with the average linkage method (version 3.1.1)[100]. The UpSet plot was generated with the UpSetR package (version1.4.0)[101]. Functional enrichment was performed using Metascape 3.5[102] and validated by orthogonal approaches using DAVID (version 6.8)[103] and Panther (version 14.1)[104], employing default parameters. Gene Ontology (GO) plots were created using the ggplot2 package (version 3.3.2).

Overlap with DamID targets of Nkx2-5 in mouse HL-1 cells (GSE44902)[52] was performed by first retrieving the zebrafish orthologues of the EnsEMBL IDs of the mouse DamID targets using BioMart (release 100)[53]. Then, the intersection of the EnsEMBL IDs for the zebrafish DamID targets and zebrafish DEGs were obtained with BioVenn (version 1.1.3)[105]. Network reconstructions were performed by retrieving protein-protein interacting relationships using STRING (version 11.0, default parameters, https://string-db.org) and visualized with Cytoscape (version 3.8.0)[106]. For networks associated with GO terms, interactions only between genes associated with specified GO terms were examined. However, the networks highlighting DamID candidates as hubs (Fig. 6) were reconstructed based on an overlap of all targets and sets A and B (Fig. 5). The first neighbors of the DamID targets and modules with more than seven nodes were included for analysis.

**Epicardial explant culture.** Heart explant cultures were performed as previously described using *Tg(tcf21:DsRed2)$^{pd37}$* fish[65]. Briefly, hearts were extracted at three days post-amputation (dpa) from wild-type and *nkx2.5$^{-/-}$;Tg(hsp70l:nkx2.5-EGFP)* fish. The ventricular apices were cultured in 24-well plates, pre-coated with fibrin, and incubated for four days at 28 °C, 5% $CO_2$. Cell migration was assessed using an EVOS Digital Microscope with ImageJ to measure the maximal radius between the edge of the ventricular apex and the edge of the monolayer. For EdU incorporation experiments, 25 mM EdU was added to the culture medium and the explants were fixed one hour later[107]. The Click-iT EdU Imaging Kit (Invitrogen) was employed following the manufacturer's instructions.

**Swimming endurance assay.** Adult 4-month-old zebrafish were kept unfed for 24 hours prior to initiation of the challenge in the swim tunnel respirometer device (Mini Swim-170, Loligo Systems). After a minute of acclimation inside the tunnel with no current, water (26 °C) flow speed was initiated at 10 cm/s and increased by 5 cm/s increments every minute. Once the maximum speed was attained, fish were allowed to swim until exhaustion or the end of the test (maximum duration of 35 min)[33].

**Reporting summary.** Further information on research design is available in the Nature Research Reporting Summary linked to this article.

## Data availability

Previously published datasets employed in this manuscript include the zebrafish reference genome GRCz10-91 (https://www.ncbi.nlm.nih.gov/assembly/GCF_000002035.5/) and Nkx2-5 DamID targets under accession number GSE44902. The RNA sequencing data newly reported in this paper is available under accession number GSE164966. Source data are provided with this paper. All other relevant data may be requested from the corresponding author.

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

## Acknowledgements

We thank Kenneth D. Poss, Caroline E. Burns, C. Geoffrey Burns, and Nadia Mercader for sharing protocols and Ian C. Scott and Enzo R. Porrello for their critical reading of the manuscript. We appreciate the support of Traude Beilharz for the bioinformatic data analysis involved in this project and the expert advice of Jingli Cao regarding techniques for the study of zebrafish cardiac regeneration. We are grateful to the Targoff Laboratory for constructive feedback and to Joshua Barber for his expert zebrafish care. We also thank the Confocal and Specialized Microscopy Shared Resource of the Herbert Irving Comprehensive Cancer Center at Columbia University Irving Medical Center, the Stem Cell Core Facility of the Columbia Stem Cell Initiative, and the Comparative Pathology Unit, the Bioimaging Unit, and the Zebrafish Unit at the Instituto de Medicina Molecular João Lobo Antunes (iMM-JLA). This work was support by grants to K.L.T. from the National Institutes of Health (R01 HL131438-01A1), C.d.S.-T. from Fundação para a Ciência e Tecnologia (FCT) (PTDC/BIA-BID/28572/2017), and M.R. from the State Government of Victoria and the Australian Government for the Australian Regenerative Medicine Institute and by Novo Nordisk Foundation (NNF21CC0073729).

## Author contributions

C.d.S.-T. and K.L.T. conceived of the project and designed the experiments. C.d.S.-T., A.G.A., C.F, D.Y., and J.K.H. optimized the rescue experiments necessary to create the *nkx2.5*$^{-/-}$ fish, generated data, and produced figures. A.V. and M.R. performed bioinformatic analysis. K.L.T provided funding for the project. C.d.S.-T., A.V., L.S., M.R., and K.L.T. wrote, reviewed, and edited the manuscript.

## Competing interests

The authors declare no competing interests.
