## [Peer Review File · Nature Communications]

Activation of Nkx2.5 Transcriptional Program is Required for Adult Myocardial RepaireREVIEWER COMMENTS

Reviewer #1 (Remarks to the Author):

In this manuscript, the authors tested whether Nkx2.5 regulates cardiac regeneration in the zebrafish model. The group showed in a previous publication that forced transgenic heat-shock driven expression of Nkx2.5 at 21 somites is sufficient to rescue embryonic lethality and allow the fish to grow to adulthood, suggesting that Nkx2.5 does not have a major role in cardiac homeostasis beyond embryogenesis. This system allows them to raise mutant adult fish and test if they are defective in cardiac regeneration following ventricular resection. They present data suggesting that there are transcriptional and regenerative defects in the mutant animals and identify from transcript profiling data several candidate contributing features. There are however major weaknesses in the study.

1) The authors claim that the adult hearts in the rescued animals are normal (as they already suggested from the previous publication). However, the sections presented show major differences in the WT, TG-WT, and TG-MT samples. For example, the mutant hearts appear much larger, rounder, and myocardium far less dense. This is perhaps because the sections are not well matched. This is a confounding problem throughout the study, which relies largely on presentation of selected sections. If they are not precisely matched it is very challenging to interpret the results. Better would be to have several sections across the heart.

2) The authors claim that Nkx2.5 is re-expressed (or as in the title, activated) upon resection. However, this is not demonstrated. The ISH never compares expression at day 0, so its impossible to interpret if it is "activated" at day 4. Controls such as a sense strand probe are missing. The equivalent experiments using a transgenic reporter are not at all convincing of re-expression or activation.

3) Much of the paper reports bioinformatics of transcript profiles comparing for example WT, transgenic WT (TG-WT) or transgenic mutant (TG-MT) heart tissues taken either before or following resection. However, all three groups show highly divergent expression patterns, such that as many genes (thousands) are differentially expressed (DEGs) comparing the two WT cohorts as across to mutants. This indicates very significant noise in the datasets and makes it very difficult to judge what is important for the mutant conditions. For example, TG-MT vs TG-WT and TG-MT vs WT show more differences than similarities. PCA analysis might be useful to know how well these samples are actually matched.

4) Most importantly, the authors suggest that Nkx2.5 "is required for adult myocardial repair" (quoting from the title). This is obviously not true, since the hearts repair very well. The only defects shown are at day 50 and in what is presumably a representative "severe" example, very minor scarring is demonstrated. Here the figures suffer also from poorly matched sections. Again, transcript patterns are just as diverse comparing WT cohorts. It is interesting if there is a minor defect, but the significance is very questionable, especially with low sample numbers.

5) Defects in cell proliferation and expression of embCMHC expression (Fig. 7) are not convincing. The authors present bar graphs indicating statistical significance, but the raw data shown is not convincing and again is subject to field selection. How was %PCNA cells actually determined?

6) A large amount of network analysis generates some observations of possible biological features but it is unclear if any of this is biologically significant and no mechanistic hypothesis was tested.

7) Defects in epicardial "integration" are also not convincing. There appear to be many more tbx18+ cells in the WT section shown compared to the MT section. Yet RNA-seq showed no difference in expression levels, suggesting again issues in field selection.

Reviewer #2 (Remarks to the Author):

In this manuscript the authors identify a role for nkx2.5 during zebrafish heart regeneration. They use a rescue model that they previously developed to overcome the embryonic lethality of knocking out nkx2.5, allowing to analyse the loss of nkx2.5 in the adult heart. Rescued hearts are morphologically normal, but show a difference in gene expression expected with loss of Nkx2.5 using RNAseq. Ventricular resection shows reduced regeneration in the rescued hearts which seems largely due to reduced myocardial proliferation as shown by IHC and RNAseq analysis. Additionally, as nkx2.5 is only expressed in the cardiomyocytes, they do not observe reduced epicardial migration and proliferation in explants of the rescued hearts.

As nkx2.5 is an important transcription factor during heart development, this study and the identification of a role for nkx2.5 during heart regeneration is definitely of interest to the field. However, this study does not reveal much more than just that and is in my opinion not suitable for a journal like nature communications. The lack of mechanistic data other than from RNAseq makes this study a better fit for a more specialized journal.

Major concerns:

- Figure 2: The authors deduce that overexpression of nkx2.5 at 20 somites leads to non-specific variation in adult myocardium. The number of DE genes is of a similar level between WT0 and TG0 as it is between TG0 and MT0. The difference between WT0 and TG0 seems very high as this is only supposed to be a very short overexpression of nkx2.5 at 20 somites and needs to be addressed further. Is EGFP or nkx2.5 still high in the adult TG0 hearts? The difference seems bigger than just 'housekeeping genes' and needs to be explained to be able to understand the model and all following experiments.
- The RNAseq experiments (uninjured and injured) show interesting gene networks, but these are not further explored. DE expression of some of these genes needs to be shown on sections and for a journal like nat comms, mechanistic insight needs to be shown, which is completely lacking.
- Bmp4 is identified as central connector, but not even expression of bmp4 in the rescued hearts is validated by ISH.
- Figure 4: Scarring in the rescued hearts seems all around the ventricle, is this observed in all hearts? The genes analysed by qPCR in this figure seem mostly ECM related genes. A difference in these genes is expected as there is more scar. It would be much more interesting to see validation of for example more muscle structure development genes. Nrg1 and twist1b seem strongly differentially expressed, this could also further be examined.
- Figure 8 (+3): Nkx2.5 is shown to be expressed in the myocardium and not overlapping with raldh2, what was the rationale to look for an epicardial defect? It would have made more sense to further focus on the myocardial differences. Tbx18 and tcf21 expression seems reduced in the rescued hearts on sections, but this is not quantified. Expression of these genes in the RNAseq data (L) seems higher though? No epicardial migration and proliferation defect is observed though and this data does not clarify the observed phenotype.

Reviewer #3 (Remarks to the Author):

In the manuscript from de Sena-Tomás et al, the authors examine the role of Nkx2.5 in heart regeneration of adult zebrafish. The authors are able to raise mutant fish to adulthood through the use of a heat-shock inducible nkx2.5-GFP transgene. Surprisingly, despite only receiving nkx2.5 during late somitogenesis, the fish survive well and have no overt signs of defects in cardiomyocyte structure within the hearts. However, some of the cardiomyocytes in the ventricle express amhc, an atrial-

specific myosin. Comparisons are made between RNA-seq of the ventricular apices from WT, transgenic, and rescued mutants with the transgene. The rescued nkx2.5 mutant hearts show defects in embryonic pathways. With respect to injury, the authors find that nkx2.5 mutants do not fully resolve the wound and at least through 50 dpa they have larger scars than the WT or transgenic fish. RNA-seq supports that defects in nkx2.5 mutant injured hearts have a failure in cell cycle and proteolysis genes. They find that in the injured hearts from nkx2.5 mutants the cardiomyocytes in the border zone do not proliferate as well nor do they express embryonic myosin, corroborating the transcriptomic analysis. The authors find that loss of nkx2.5 also has potential non-autonomous effects as the epicardium does not invade the injured area in the nkx2.5 mutant hearts.

Overall, this is a very interesting study of a unique tool that provides new insight into regeneration in zebrafish. The observations are important in that Nkx2-5 mutations are found in a significant portion of congenital heart defects. Thus, the information in the paper could provide the foundation for both inroads into understanding vertebrate regeneration and potential therapeutic strategies for congenital heart defects in patients with Nkx2-5 mutations. The information in the manuscript will be of interest to the cardiac development and regeneration communities.

The statistical analysis appears to be appropriate for the experiments performed.

While I am generally enthusiastic about the fundamental information provided in the manuscript, I think there are some key areas where it could be improved.

Major issues:

- 1) It is stated that myocardial architecture is normal in the hearts. However, can the authors clarify this? Is there any difference in the size of the nkx2.5 mutant hearts? The images used in Fig. 1 seem to indicate that the mutant heart is larger than the controls. Is this just due to the specific heart and sections or actually reflect a difference in the mutant hearts compared to WT and transgenic fish?
- 2) While there are defects in embryonic gene profiles, what does this look like in the heart? Were some of these differentially expressed genes in Fig. 2 confirmed with in situ hybridization? Do these genes show specific localization to the cardiomyocytes? With regard to the interpretation, what does it mean that there are defects in embryonic pathways, but the hearts are essentially normal? Does it mean those pathways are not required for normal homeostasis or cardiomyocyte maturation in adults?
- 3) The reactivation of Nkx2.5 is a bit difficult to see in Fig. 3 with the in situ, due to the lack of cellular architecture visible. The expression of Nkx2.5 in re-differentiating cardiomyocytes in the wound is more evident with the transgene. There is a commercially available Nkx2.5 antibody. Was that used to examine Nkx2.5 in the resected hearts? Did it show the same types of localization as the nkx2.5:yellow transgene?
- 4) While the response of the transgene to the resection injury is only about a day, it could be argued this is not insignificant. Where was the transgene activated? Was it along the border zone? Can a GFP antibody be used to determine where it is activated within the injured heart?
- 5) It is not clear what the rationale for going straight to 50 dpa when examining the wounds. Does this mean that the nkx2.5 mutant hearts never fully heal or are they just delayed? What do the wounds look like at some intermediate stages of regeneration?
- 6) While the nkx2.5 mutants fail to express embryonic myosin, do they fail to express other indicators of de-differentiation, such as Gata4? The effects on the de-differentiation process could be a little more thoroughly examined. Is the regeneration process completely blocked in the nkx2.5 mutants or just slightly impaired, as the scar area, while still evident, seems much smaller compared to if regeneration was not occurring at all? In many of the images, while the defects on the cellular

markers is evident, it isn't clear the size of the wound area is that different.

7) With respect to the epicardial response, the *tcf21:dsred* transgene is convincing. However, the *tbx18* defect is difficult to discern from the images. It looks as though the wound is just smaller in those embryos, not so much that there is a failure of the *tbx18* cells to reintegrate. Can the *tbx18* relative to the wound be used or different representative images with quantification be incorporated?

8) While the bioinformatic comparisons are very important, the descriptions of that aspect of the study is very dense. In general, it would seem that the description of the bioinformatics could be streamlined quite a bit more and integrated a bit more smoothly with the experimental portions of the paper. Aspects of the bioinformation description are also a bit confusing in places, such as in Fig 4 where the TG is being compared to the WT. In the confirmation with the RT-PCR, the mutant comparison is then stuck in there too without much rationale. Additionally, there is a lot of speculation about the functions of some of these pathways that are identified with respect to specific regulators that are not backed up with experimental data. The findings can certainly be used as rationale for the some of the experiments. However, the speculation seems a bit excessive in parts and one of the aspects that weighs down the middle of the paper. For instance, the detailed coverage about *ect2* or the proteolysis genes in other settings seems more appropriate for the discussion. It is not clear as presented now that they should be in the results.

Minor issues:

1) It is interesting that the adult mutant hearts also have *amhc* expression in the ventricle. Do these cardiomyocytes co-express atrial and ventricular genes (at least *amhc* and *vmhc*) or do they only express atrial cardiomyocyte markers?

2) While the incorporation of the transgenic control is necessary and stated it is not leaky, is there evidence that its reactivation has any gain-of-function effect? While possibly beyond the scope of the paper, could a cryoinjury, which induces prolonged activation of the inducible *nkx2.5* be used to test if increased *nkx2.5* is sufficient to promote regeneration.

3) The network diagrams/pathways in the some of the figures could be moved to the supplemental and/or reduced in the main figures. It is not clear that how they are displayed with the networks is all that critical for the information being portrayed. Additionally, the fonts for the scales for the fold change are so small they are very difficult to read.

4) For Fig. 4E are the shades correct? Should Moderate be dark gray and severe be black?

5) It is stated that the transgenic mutants will just be referred to as *nkx2.5*^{-/-}. Most of the figures include the name of the transgene too, which is likely not necessary.

Response to Reviewers – de Sena-Tomás et al. (NCOMMS-20-47176)

We are grateful for the thoughtful and constructive comments provided by the reviewers and appreciate the feedback that our manuscript represents “*a very interesting study of a unique tool that provides new insight into regeneration in zebrafish*”. Furthermore, the reviewers were “*enthusiastic about the fundamental information provided*” and deemed the observations as “*important*” to provide “*inroads into understanding vertebrate regeneration and potential therapeutic strategies for congenital heart defects*”.

Moreover, the comments provided by each reviewer prompted significant revisions to our manuscript and we believe these modifications have substantially improved our study. As a consequence of these efforts, our revised submission has been meaningfully delayed due to COVID-19 restrictions and the extensive experimental design required to produce adult *nkx2.5* mutant fish to generate additional data. With these results in hand now, we have comprehensively responded to each conceptual concern and experimental question below.

Reviewer #1

1) The authors claim that the adult hearts in the rescued animals are normal (as they already suggested from the previous publication). However, the sections presented show major differences in the WT, TG-WT, and TG-MT samples. For example, the mutant hearts appear much larger, rounder, and myocardium far less dense. This is perhaps because the sections are not well matched. This is a confounding problem throughout the study, which relies largely on presentation of selected sections. If they are not precisely matched it is very challenging to interpret the results. Better would be to have several sections across the heart.

We are grateful for this input and agree that it is challenging to compare tissue architecture and morphology from sections with variability. To address this concern, we performed additional experiments in WT0, TG0, and MT0 hearts and generated a figure with analogous sections in each genotype (new Fig. 1). Moreover, to dissect the differences observed in the original Fig. 1, we executed high resolution imaging to capture z-disk structure and identified organized and prominent striations indicating intact sarcomeres in the WT0, TG0, and MT0 hearts (new Fig. 1N-P). Next, we quantified ventricular volume and area of trabecular tissue. While the MT0 hearts have a slight increase in ventricular volume compared to WT0 and TG0 (new Fig. 1R), a finding consistent our prior report (George et al., *Developmental Biology*, 2015), the extent of trabecular network in the ventricles remains unchanged (new Fig. 1Q). We then assessed the fitness of the WT0, TG0, and MT0 fish to determine whether this minor discrepancy in ventricular size contributed to a defect in cardiac function. Through swim tunnel assessments (new Fig. 1S), we observed no differences between the three genotypes in stamina duration. Taken together, we conclude that, while *nkx2.5*^{-/-} fish

exhibit subtle variation in ventricular size that warrants further investigation, no gross morphological or functional deficits in the adult *nkx2.5* loss-of-function model are evident.

2) The authors claim that Nkx2.5 is re-expressed (or as in the title, activated) upon resection. However, this is not demonstrated. The ISH never compares expression at day 0, so it's impossible to interpret if it is "activated" at day 4. Controls such as a sense strand probe are missing. The equivalent experiments using a transgenic reporter are not at all convincing of re-expression or activation.

We thank Reviewer #1 for the valuable feedback and have updated Fig. 3 to include uninjured specimens. In addition, we repeated the colorimetric RNAscope experiments assessing *nkx2.5* expression employing fluorescent RNAscope strategies for improved resolution. We present these data in a new version of Fig. 3 where we demonstrate that *nkx2.5* is normally observed in the uninjured myocardium. At 7 dpa, a few *nkx2.5* transcripts are identified at the site of the resolving injury. By 14 dpa, increased pockets of *nkx2.5* expression are apparent in the regenerate. From these results, we conclude that initiation of the *nkx2.5* gene regulatory network normally occurs during regeneration. To reinforce this conclusion, we left the images illustrating the *Tg(nkx2.5:ZsYellow)* expression given positive feedback regarding these data from Reviewer #3 (point 3).

3) Much of the paper reports bioinformatics of transcript profiles comparing for example WT, transgenic WT (TG-WT) or transgenic mutant (TG-MT) heart tissues taken either before or following resection. However, all three groups show highly divergent expression patterns, such that as many genes (thousands) are differentially expressed (DEGs) comparing the two WT cohorts as across to mutants. This indicates very significant noise in the datasets and makes it very difficult to judge what is important for the mutant conditions. For example, TG-MT vs TG-WT and TG-MT vs WT show more differences than similarities. PCA analysis might be useful to know how well these samples are matched.

We indeed observed a high number of DEGs between comparisons from each genotype before or following resection. In order to explain these differences and their contribution to the phenotype, as per this Reviewer's request, we performed PCA analysis to compare all samples and incorporated this graph into our new Fig. S6. First, the PCA shows that duplicates for each condition cluster together, indicating that the DEGs are not due to noise. Second, the first dimension tends to separate injured versus uninjured samples, while, the second dimension tends to separate samples by genotype (WT, TG, and MT). Taken together, these data indicate a robust, global transcriptional difference in MT versus WT or TG samples. We did, however, observe a larger separation between the WT0 and TG0 samples (in comparison with TG7 and WT7 samples), which we have resolved to be secondary to differences in 'housekeeping' functions (new Fig. S2A) (see also answer to Reviewer #2, point 1). This distribution is reinforced by the graph below (which is included now in new Fig. S1). Moreover, we account for this variation in our

analysis by removing the contribution of these DEGs to the WT0 and TG0 comparison by generating the 'overlap' set (new Fig. 2A, group C).

4) Most importantly, the authors suggest that *Nkx2.5* “is required for adult myocardial repair” (quoting from the title). This is obviously not true, since the hearts repair very well. The only defects shown are at day 50 and in what is presumably a representative “severe” example, very minor scarring is demonstrated. Here the figures suffer also from poorly matched sections. Again, transcript patterns are just as diverse comparing WT cohorts. It is interesting if there is a minor defect, but the significance is very questionable, especially with low sample numbers.

We appreciate Reviewer #1’s concern that the *nkx2.5*^{-/-} hearts repair well given our presentation of images at 50 dpa. However, we chose this time point to emphasize that there is an extended delay in the healing process. Furthermore, we did not select the most severe examples in the original draft of the manuscript as we were eager to illustrate the range of severity. Yet, to emphasize the impressive nature of the scar in the *nkx2.5*^{-/-} compared to the non-transgenic wild-type and transgenic fish, we repeated amputation experiments to compare WT, TG, and MT at 30 dpa and 50 dpa. We have updated Fig. 4 to include these data demonstrating the dramatic impairment in scar resolution in the MT compared to the WT or TG hearts at 30 dpa. Furthermore, we incorporate quantification of residual scar and scar severity from the 30 dpa experiments into the new version of Fig. 4. In our supplemental data (Figs. S3 and S4), we present additional examples of 30 dpa and 50 dpa hearts in all genotypes to underscore our critical finding that *Nkx2.5* is required for adult myocardial repair. Altogether, our new data provide added support to this conclusion by bolstering our sample numbers too.

5) Defects in cell proliferation and expression of *embCMHC* expression (Fig. 7) are not convincing. The authors present bar graphs indicating statistical significance, but the raw data shown is not convincing and again is subject to field selection. How was %PCNA cells determined?

We acknowledge the subtle nature of the data demonstrating proliferation and *embCMHC* expression (original Fig. 7). Thus, we have updated this figure to represent better the abundance of proliferative cells in WT7 compared to MT7 hearts (new Fig. 8A-D). We also present higher magnification views to highlight the visibility of the increased number of PCNA⁺ cells in the WT7 hearts and represent more accurately the quantitative data that we acquired. Furthermore, we improved the clarity of our description of the technique

by which we calculated the percentage of PCNA⁺ cardiomyocytes in the Quantification and Statistical Analysis section of Materials and Methods. For the embCMHC expression, we also incorporated higher magnification views in individual channels to reinforce the dramatically diminished expression of this dedifferentiation marker in the *nkx2.5*^{-/-} hearts. Finally, we performed an additional experiment employing Alcam to substantiate our results. In new Fig. S10, we show that MT7 myocardium exhibits a statistically significant dampening of Alcam expression when compared to WT7, validating decreased dedifferentiation in the MT7 injured zone.

6) A large amount of network analysis generates some observations of possible biological features, but it is unclear if any of this is biologically significant and no mechanistic hypothesis was tested.

We understand Reviewer #1's point regarding biological significance and would like to underscore the relevance of our findings and our additional efforts to investigate novel effectors downstream of Nkx2.5. In this manuscript, we hypothesize that, following injury of the adult heart, Nkx2.5 is necessary to stimulate essential cell cycle genes, repress mitochondrial metabolism gene network, and maintain proteolysis mechanisms for the induction of dedifferentiation and proliferation of cardiomyocytes in the regenerate. To test these mechanisms, we performed new RNAscope and *in situ* hybridization experiments to underscore the function of specific candidates presented in this network analysis. Specifically, in new Fig. 7, these data emphasize the depletion of *ect2* expression in MT7 compared to WT7 injuries supporting its role as a Nkx2.5 effector in promoting proliferation following injury. Furthermore, statistically significant reduction in *psmb3* and *psmd7* expression in MT7 compared to WT7 authenticates the importance of these targets in maintaining the dedifferentiation program regulated by Nkx2.5.

*7) Defects in epicardial "integration" are also not convincing. There appear to be many more *tbx18*⁺ cells in the WT section shown compared to the MT section. Yet RNA-seq showed no difference in expression levels, suggesting again issues in field selection.*

We realize discrepancy in field selection of the *tbx18* ISH in the original Fig. 8 makes it difficult to visualize the epicardial integration defect in the *nkx2.5*^{-/-} regenerating myocardium. In our experience, we observe variability with *tbx18* expression following injury, creating challenges in correlating its expression pattern and RNA-seq data. Thus, we decided to omit these results and enhance our analysis of the *Tg(tcf21:dsRed)* expression pattern through quantification methods, as presented in the new Fig. 9E. Specifically, we measured the distance from the apices of the injured hearts to the farthest point of EPDC penetration and found statistically significant decrease in migration distances between WT and MT at 14 dpa. Taken together, these data reinforce our conclusion that the failure of epicardial integration is a result of non-cell autonomous defects in the *nkx2.5*^{-/-} microenvironment such as essential extracellular matrix (ECM) factors functioning downstream of Nkx2.5 (Fig. 5C).

Reviewer #2

1) *Figure 2: The authors deduce that overexpression of nkx2.5 at 20 somites leads to non-specific variation in adult myocardium. The number of DE genes is similar between WT0 and TG0 as it is between TG0 and MT0. The difference between WT0 and TG0 seems very high as this is only supposed to be a very short overexpression of nkx2.5 at 20 somites and needs to be addressed further. Is EGFP or nkx2.5 still high in the adult TG0 hearts? The difference seems bigger than just 'housekeeping genes' and needs to be explained to be able to understand the model and all following experiments.*

Similar to point 3 by Review #1, we concur that there is a large number of DEGs between the WT0 and TG0. In order to address this finding further, we performed quantitative qPCR analysis to determine if the *nkx2.5* expression is higher in the adult TG0 compared to WT0 hearts. Our results demonstrate that there is no statistically significant increase in the endogenous transgenic expression in the TG0 compared with the WT0 (new Fig. S1B). Therefore, we conclude that the discrepancy in this dataset is not due to the presence of *Tg(hsp70l:nkx2.5-EGFP)*. Given that the most highly enriched GO terms in the TG0 vs WT0 comparison (new Fig. 2A, group D) are autophagy, mitochondrion organization, and ribosome biogenesis, our data suggest that aberrations in these 'housekeeping' functions constitute the primary differences in this comparison and are distinct from the molecular changes observed in the MT0 comparisons. While 'housekeeping' functions represent >75% of the DEGs in the TG0 vs WT0 comparison, these GO terms compose only ~5% of the DEGs in the MT0 comparisons (new Fig. S1C). We implemented measures to remove confounding variables secondary to these alterations by eliminating the contribution of DEGs between WT0 and TG0 in the 'overlap' set (new Fig. 2A, group C). This strategy allowed us to isolate genetic differences attributable to the loss of *nkx2.5* gene function in the adult heart.

2) *The RNAseq experiments (uninjured and injured) show interesting gene networks, but these are not further explored. DE expression of some of these genes needs to be shown on sections and for a journal like nat comms, mechanistic insight needs to be shown, which is completely lacking. Bmp4 is identified as central connector, but not even expression of bmp4 in the rescued hearts is validated by ISH.*

We appreciate Reviewer #2's concern about the need to clarify mechanistic insight and demonstrate expression patterns of candidate DEGs. To this end, we performed several new experiments. While we had difficulty with the *bmp4* probe, we selected other central connectors for *in vivo* validation and illustration of mechanisms mediated by Nkx2.5. In the new Fig. 2, we employed RNAscope to illustrate upregulation of *vcana* in MT0 versus WT0 as validation of the differences illustrated in the embryonic GRN between these two genotypes. Furthermore, as discussed further in response to Reviewer #1's point 6, our new data exhibits decreased expression of *psmb3* and *psmd7*, critical nodes involved in the proteolysis network, in the *nkx2.5*^{-/-} hearts following injury as evidence of the mechanistic requirement of Nkx2.5 in maintaining a dedifferentiation program to facilitate

regeneration (new Fig. 7A-D). Finally, diminished *ect* expression in the regenerate of MT7 versus WT7 hearts following amputation supports our conclusion that Nkx2.5 is required to regulate *ect2* to boost proliferation (new Fig. 7E,F).

3) Figure 4: Scarring in the rescued hearts seems all around the ventricle, is this observed in all hearts? The genes analysed by qPCR in this figure seem mostly ECM related genes. A difference in these genes is expected as there is more scar. It would be much more interesting to see validation of for example more muscle structure development genes. Nrg1 and twist1b seem strongly differentially expressed, this could also further be examined.

We agree that the image in the original Fig. 4 is misleading as we do not observe scarring around the ventricle in all rescued hearts. To accentuate our findings more precisely, we present newly-acquired images demonstrating the focal nature of the scar in the *nkx2.5*^{-/-} heart in an updated version of Fig. 4, with additional examples in Figs. S3 and S4. We also include the percentage of scar tissue and the quantitative analyses of scar severity (new Fig. 4E,F). Furthermore, we appreciate the suggestion to perform validation of ‘muscle structure development genes’ and have extended the qPCR verification of the RNA-seq data by including *cacna1c*, *pdlim5b*, *popdc2*, and *tcap*. In total, out of 14 genes assayed (new Fig. S6D,E), 11 were similarly up or downregulated by qPCR. Specifically, three out of four ‘muscle structure development genes’ matched the RNA-seq data in trend. Furthermore, while these findings corroborate the accuracy of our wild-type sequencing data, we acknowledge that subtle changes in the border zone may be difficult to capture as qPCR was performed on ventricular apical tissue. Yet, taken together, our data demonstrate that the intersection of the regenerative responses in both control systems reflects previously identified cellular repair processes and that our RNA-seq transcriptional profiles echo *in vivo* alterations in transcriptional regulation.

4) Figure 8 (+3): Nkx2.5 is shown to be expressed in the myocardium and not overlapping with raldh2, what was the rationale to look for an epicardial defect? It would have made more sense to further focus on the myocardial differences. Tbx18 and tcf21 expression seems reduced in the rescued hearts on sections, but this is not quantified. Expression of these genes in the RNAseq data (L) seems higher though? No epicardial migration and proliferation defect is observed though and this data does not clarify the observed phenotype.

We thank Reviewer #2 for this thoughtful recommendation to clarify our rationale in examining potential endocardial and epicardial defects in the *nkx2.5*^{-/-} fish. Although we did not observe expression of Nkx2.5 in the endocardium or epicardium, our data revealed misregulation in ECM genes and this insight prompted further investigation of putative non-cell autonomous functions of Nkx2.5. Thus, in addition to examining the cell autonomous roles of Nkx2.5 in the myocardium, such as promoting proliferation and dedifferentiation (new Fig. 8), we scrutinized potential aberrant mechanisms involving other cell types, such the endocardium and epicardium. We detected no differences

between wild-type and *nkx2.5*^{-/-} hearts in angiogenic sprouting (new Fig. S11). Yet, we discerned impaired epicardial integration (new Fig. 9) in the *nkx2.5*^{-/-} compared to wild-type fish. In light of these findings, our data in explant cultures validate our hypothesis that the defective epicardium is not responsible for the diminished penetration in the regenerate. Instead, non-cell autonomous roles of *nkx2.5* explain this finding and underline the significance of Nkx2.5 in mediating regenerative potential extrinsic to cardiomyocytes via modulation of ECM (Fig. 5C).

In order to strength our representation of the epicardial defect, we implemented quantitative morphometrics to measure the distance of penetration of *Tg(tcf21:dsRed)*⁺ cells in the regenerate between the WT7 and MT7. Our analysis illuminates statistically significant impairment in the epicardial migration in the absence of *nkx2.5* (new Fig. 9E). Finally, we removed the quantitative RNA-seq data from the original Fig. 8 as correlation of transcript levels and epicardial patterning is challenging due to the different modalities used to assay the data and also the complications of field selection.

Reviewer #3

Major issues:

1) It is stated that myocardial architecture is normal in the hearts. However, can the authors clarify this? Is there any difference in the size of the nkx2.5 mutant hearts? The images used in Fig. 1 seem to indicate that the mutant heart is larger than the controls. Is this just due to the specific heart and sections or actually reflect a difference in the mutant hearts compared to WT and transgenic fish?

We thank Reviewer #3 for emphasizing this point and prompting further characterization of WT0 compared to TG0 and MT0 hearts. From newly-generated tissue specimens and enhanced quantitative morphometric analysis (new Fig. 1), we present data demonstrating that the MT0 ventricles are slightly larger in volume than WT0 or TG0 hearts (new Fig. 1R), as previously noted (George et al., *Developmental Biology*, 2015). However, our studies do not reveal significant differences in the ventricular architecture as assessed by the proportion of trabeculation or the z-disk organization when comparing each genotype (new Fig. 1N-Q). Furthermore, employing swim tunnel challenges, we show that the subtle cardiomegaly does not result in deficits in oxygen carrying capacity or cardiac fitness between MT0 and WT0 fish (new Fig. 1S). Although the slight alteration in ventricular volume warrants further investigation, these findings underscore the crucial conclusion that there are no significant differences in tissue morphology or cardiovascular capacity in the *nkx2.5* loss-of-function model when compared to wild-type.

2) While there are defects in embryonic gene profiles, what does this look like in the heart? Were some of these differentially expressed genes in Fig. 2 confirmed with in situ hybridization? Do these genes show specific localization to the cardiomyocytes? With regard to the interpretation, what does it mean that there are defects in embryonic

pathways, but the hearts are essentially normal? Does it mean those pathways are not required for normal homeostasis or cardiomyocyte maturation in adults?

We appreciate this comment and, in response, have performed additional *in situ* hybridization experiments that are included in the new version of Fig. 2 to underscore the correlation between the perturbed embryonic GRNs and expression profiles. Specifically, we demonstrate upregulation of *vcana* in MT0 compared with WT0 using RNAScope (new Fig. 2C,D).

Regarding our interpretation of these data, we believe that there are adequate compensatory mechanisms to buttress the processes of cardiomyocyte maturation in the context of this disrupted embryonic pathways in the *nkx2.5* loss-of-function model. But, in the context of myocardial injury, these compensatory mechanisms are inadequate to ensure an appropriate reparative response in the *nkx2.5*^{-/-} heart.

3) The reactivation of Nkx2.5 is a bit difficult to see in Fig. 3 with the in situ, due to the lack of cellular architecture visible. The expression of Nkx2.5 in re-differentiating cardiomyocytes in the wound is more evident with the transgene. There is a commercially available Nkx2.5 antibody. Was that used to examine Nkx2.5 in the resected hearts? Did it show the same types of localization as the nkx2.5:zsyellow transgene?

We agree that the cellular architecture is not ideal in the images originally presented in Fig. 3. Thus, we repeated these assays with RNAScope to improve visualization; our data are featured in our new version of Fig. 3. We believe that these findings are more convincing and verify our conclusion that *nkx2.5* is expressed throughout the myocardium in the uninjured adult heart (new Fig. 3A). Moreover, at 7 dpa and 14 dpa, progressively increased *nkx2.5* expression is evident in the regenerate (new Fig. 3B,C), substantiating our conclusion that Nkx2.5 is required for dedifferentiation and proliferation during myocardial regeneration. These findings are further clarified in response to point 2 by Reviewer #1. Finally, while there is a commercially available Nkx2.5 antibody, we did not have success using this reagent for immunostaining in the adult heart.

4) While the response of the transgene to the resection injury is only about a day, it could be argued this is not insignificant. Where was the transgene activated? Was it along the border zone? Can a GFP antibody be used to determine where it is activated within the injured heart?

We recognize the concern presented by Reviewer #3 regarding the spatiotemporal activation of the *Tg(hsp70l:nkx2.5-EGFP)* following amputation. To probe this question, we performed new immunostaining experiments employing Tropomyosin and anti-GFP antibody to detect expression of Nkx2.5-EGFP over time (new Fig. S2). Our data uncover parallel, albeit slightly delayed in comparison to qPCR evaluation of *nkx2.5-EGFP* transcripts, expression of Nkx2.5-EGFP with diminution of the protein by 2 dpa. Nkx2.5-EGFP is evident within ventricular myocardium and not specifically at the injury border. Moreover, Nkx2.5-EGFP expression is cytoplasmic initially (new Fig. S2A-C) and

becomes localized to the nucleus at 1 dpa (new Fig. S2D) prior to exhibiting downregulation at 2 dpa (new Fig. S2E). These data suggest that, while *Tg(hsp70l:nkx2.5-EGFP)* is activated in response to ventricular resection, *nkx2.5* expression is limited in duration and location, thus, minimizing the likelihood that these secondary effects modulate the response to injury.

5) It is not clear what the rationale for going straight to 50 dpa when examining the wounds. Does this mean that the nkx2.5 mutant hearts never fully heal or are they just delayed? What do the wounds look like at some intermediate stages of regeneration?

We appreciate Reviewer #3's apprehension about the decision to select 50 dpa when examining the wounds. As mentioned above in response to point 4 from Reviewer #1, to address these questions, we performed additional experiments to assess the injury in a series of wild-type and *nkx2.5*^{-/-} fish at an intermediate stage. Our new 30 dpa data in Fig. 4 authenticates the severity of the injury in the MT compared with the WT and TG, both through visualization of the AFOG staining and quantification of % scar tissue and scar severity. Moreover, we display more examples of injuries in WT, TG, and MT at 30 dpa in Fig. S3 and at 50 dpa (with equivalent morphometrics) in Fig. S4. Taken together, these findings substantiate our conclusion that *Nkx2.5* is required for myocardial regeneration and that, at a timepoint when wild-type injuries are predominantly healed, *nkx2.5*^{-/-} hearts exhibit persistently severe injuries with collagen deposition.

6) While the nkx2.5 mutants fail to expression embryonic myosin, do they fail to express other indicators of de-differentiation, such as Gata4? The effects on the de-differentiation process could be a little more thoroughly examined. Is the regeneration process completely blocked in the nkx2.5 mutants or just slightly impaired, as the scar area, while still evident, seems much smaller compared to if regeneration was not occurring at all? In many of the images, while the defects on the cellular markers is evident, it isn't clear the size of the wound area is that different.

We thank Reviewer #3 for this helpful feedback regarding investigation of the dedifferentiation defect in the *nkx2.5*^{-/-} fish. First, from our new data in Figs. S3 and S4, we have the opportunity to present the range of scars more comprehensively and to illustrate the severity of the injury at both 30 dpa and 50 dpa in the *nkx2.5* loss-of-function model. Thus, we conclude that the regeneration process is significantly blocked in the absence of *nkx2.5*. Moreover, we understand the concern that the wound area is not dramatically different between the WT7 and MT7 in our images displaying cellular markers of dedifferentiation (new Fig. 8). Yet, these data are acquired at 7 dpa and, given that minimal repair occurs by this time point, we would not expect a substantial discrepancy in the scar size. Finally, to explore the effects on dedifferentiation further, we performed immunostaining with Alcam in new Fig. S10. Our data illuminates a statistically significant inhibition of Alcam expression in the MT7 when compared to WT7, validating decreased dedifferentiation in the MT7 injury zone.

7) *With respect to the epicardial response, the tcf21:dsred transgene is convincing. However, the tbx18 defect is difficult to discern from the images. It looks as though the wound is just smaller in those embryos, not so much that there is a failure of the tbx18 cells to reintegrate. Can the tbx18 relative to the wound be used or different representative images with quantification be incorporated?*

As mentioned in response to Reviewer #1 point 7, we appreciate Reviewer #3's concern regarding the *tbx18* defect and have removed these *in situ* hybridization data given our experience with impressive temporal variability in *tbx18* expression following injury. In order to strengthen our presentation of the epicardial defect, we integrated new morphometric analysis of *Tg(tcf21:dsRed)* epicardial migration in the WT7 versus MT7 (new Fig. 9E), highlighting the statistically significant decrement in epicardial penetration in the *nkx2.5*^{-/-} myocardium.

8) *While the bioinformatic comparisons are very important, the descriptions of that aspect of the study is very dense. In general, it would seem that the description of the bioinformatics could be streamlined quite a bit more and integrated a bit more smoothly with the experimental portions of the paper. Aspects of the bioinformation description are also a bit confusing in places, such as in Fig 4 where the TG is being compared to the WT. In the confirmation with the RT-PCR, the mutant comparison is then stuck in there too without much rationale. Additionally, there is a lot of speculation about the functions of some of these pathways that are identified with respect to specific regulators that are not backed up with experimental data. The findings can certainly be used as rationale for the some of the experiments. However, the speculation seems a bit excessive in parts and one of the aspects that weighs down the middle of the paper. For instance, the detailed coverage about *ect2* or the proteolysis genes in other settings seems more appropriate for the discussion. It is not clear as presented now that they should be in the results.*

We are grateful for this advice for strategies to streamline the manuscript text. In response to these recommendations, we first removed the panels in Fig. 4 comparing TG0 to WT0 and established a new Fig. S1 to display these data. Second, we present validation of the RNA-seq data in a more cohesive manner by consolidating relevant analyses in new Fig. S6. Finally, while we toned down the speculative nature of the commentary about nodes identified through the GRN analyses, we also offer new data to support the role of these Nkx2.5 targets important for proliferation and proteolysis networks. Precisely, in new Fig. 7, our data emphasize the depletion of *ect2* expression in MT7 compared to WT7 injuries supporting its role as a Nkx2.5 effector in promoting proliferation in this context. Furthermore, statistically significant reduction in *psmb3* and *psmd7* expression in MT7 compared to WT7 authenticates the importance of these targets in maintaining the dedifferentiation program regulated by Nkx2.5.

Minor issues:

1) It is interesting that the adult mutant hearts also have amhc expression in the ventricle. Do these cardiomyocytes co-express atrial and ventricular genes (at least amhc and vmhc) or do they only express atrial cardiomyocyte markers?

While we welcome this question, we have been unable to answer it adequately given the requirement for two-color *in situ* hybridization, RNAscope, or scRNA-seq. We find this evidence of transdifferentiation in the myocardium of the *nkx2.5*^{-/-} fish fascinating and are eager to pursue underlying mechanisms in future studies. Furthermore, given that the *amhc* expression is localized to areas distal to the injury, we presume that the requirement for Nkx2.5 in identity maintenance indicates a role that is independent from its function in cardiac regeneration.

2) While the incorporation of the transgenic control is necessary and stated it is not leaky, is there evidence that its reactivation has any gain-of-function effect? While possibly beyond the scope of the paper, could a cryoinjury, which induces prolonged activation of the inducible nkx2.5 be used to test if increased nkx2.5 is sufficient to promote regeneration.

Again, we are thankful for this interesting suggestion. While we have not tested cryoinjury as a method to boost a regenerative response (and assess gain-of-function potential), we indeed performed an experiment to test whether *Tg(hsp70l:nkx2.5-EGFP)* overexpression in adults following amputation is sufficient to rescue the regenerative defect in the *nkx2.5* loss-of-function model. We executed heat shock for one hour daily for 7 days following ventricular resection in the wild-type, *Tg(hsp70l:nkx2.5-EGFP)*, and *nkx2.5*^{-/-} fish. AFOG staining revealed no statistically significant difference in the percentage of scar area between the fish that received heat shock and those that did not in each genotype category. We agree that this line of investigation requires further attention as the specific dosage and temporal kinetics of *nkx2.5* over expression may be crucial to promote sufficient pro-regenerative GRN responses.

3) The network diagrams/pathways in the some of the figures could be moved to the supplemental and/or reduced in the main figures. It is not clear that how they are displayed with the networks is all that critical for the information being portrayed. Additionally, the fonts for the scales for the fold change are so small they are very difficult to read.

We appreciate this constructive feedback and have shifted some of the network diagrams to the supplement materials. Furthermore, we have increased the font size of the scales describing fold change to enhance visibility.

4) *For Fig. 4E are the shades correct? Should Moderate be dark gray and severe be black?*

We have corrected the color shades to properly represent the appropriate phenotypic categories in new Fig. 4.

5) *It is stated that the transgenic mutants will just be referred to as nkx2.5^{-/-}. Most of the figures include the name of the transgene too, which is likely not necessary.*

Thank you for this suggestion. We have edited the figures to streamline the nomenclature employed in this manuscript.

REVIEWER COMMENTS

Reviewer #1 (Remarks to the Author):

The revised manuscript is substantially improved based on the presentation of new figures with much more convincing data. Most of my major concerns were addressed. The delay in regeneration in mutant hearts will be of general interest.

Minor points:

- 1) The manuscript would benefit from an additional round of editing for clarity. Authors are encouraged to avoid describing the study results as "exciting" "remarkable" or "innovative" as such descriptors should be left for the data to show.
- 2) Likewise terminology such as "streamline the inquiry", "eke out" or "dampened expression" could be replaced by scientifically accurate sentences. Studies published in 2017 are referred to as "recent studies".
- 3) Nowhere in this dataset are any "direct" targets of Nkx2.5 identified. If such information has been shown in other contexts it should be made clear that it is an implication based on that other data.

Reviewer #2 (Remarks to the Author):

In my original comments I had mentioned that the RNAseq experiments show interesting gene networks, but that these were not explored in enough detail, with validation missing, but also importantly any further insight that would make the paper of significant interest to the field. Validation of a number of genes is now provided, but no further mechanistic insight is given.

The provided validation on sections is confusing and does not at all support the claims. *psmb3*, *psmd7* and *ect2* seem to be expressed in the scar area, not in the myocardial wound border and cannot be directly regulated by *nkx2-5* based on this.

I am also worried about the lack of statistical difference in the qPCR validation.

As I mentioned before, the authors do have an interesting model, but as the paper is largely build around RNAseq data with no validation initially, and the data provided in response to the reviewers comments does not look convincing, I do not think the results are at all ready for publication in this journal.

Reviewer #3 (Remarks to the Author):

In the revised manuscript from de Sena-Tomás et al, the authors examine the role of Nkx2.5 in heart regeneration of adult zebrafish. I still think this is a very interesting study the employs a unique tool to provide insights into regeneration of the zebrafish heart. Although de-differentiation of cardiomyocytes is accepted to be necessary for regeneration, the role of core genes like Nkx2.5 involved in embryonic cardiomyocyte differentiation in this process has not been examined. Furthermore, Nkx2.5 mutations are found in a significant portion of congenital heart defects. Therefore, this study will be of interest to the cardiovascular development and regeneration

communities.

I remain very enthusiastic about the study performed in the revised manuscript. Furthermore, I appreciate the considerable amount of work the authors have clearly done to try to address the reviewers' comments. While I am very supportive of the manuscript and do not want to create an endless cycle of critiques and revisions for the authors, there are some areas the authors should consider changing to further strengthen the revised manuscript prior to publication.

Minor issues:

1. It would appear that in total, all the data (metabolic signatures, marker analysis, etc.) support that without Nkx2.5 the adult cardiomyocytes fail to de-differentiate properly. While this would seem to be a very succinct and important message, it is not clear to me this is ever explicitly stated and the assays are always handled somewhat independently. It would be good if the authors could simply state this to unify and convey their message.
2. The paragraph beginning on line 509 is a bit confusing. The stated goal is "to identify the molecular mechanisms underlying this defective reparative response." However, most of the paragraph is setting the stage for this analysis in the next section, which is not clearly conveyed through this paragraph, and does not deal with this actual goal. I think the logic would flow better if this paragraph were revised and moved to set up the actual comparison and defects in reparative response that is in the next section of the manuscript.
3. The background for the DamID data setting up the intersection of the different datasets is not covered beginning on line 624. This dataset was with mouse Nkx2.5 in HL-1 cells and is only mentioned in the Methods. The cross comparison of the data is of course fine. However, the rationale for the intersection and the background of the dataset should be provided to set up the experiment for the reader. Moreover, as the binding of Nkx2.5 and potential directness of the zebrafish targets is not confirmed in the context in the paper, statements regarding what the intersection of the data means, particularly in the discussion, need to be qualified.
4. There are a couple places where the references for Figure panels in the text are not in the order presented in the Figures. Fig. 2F on line 385 and Figure. 7E on line 658 are examples.
5. The *ect2*, *psmb3*, and *psmd7* confirmation in Figure 7 show that changes in expression are primarily throughout the wound and not really in the border zone. Wouldn't the expression of these genes that are proposed to be direct targets expect to be within the de-differentiating cardiomyocytes of the wild-type heart, i.e. primarily autonomous? What does this mean? On line 647, it states, " *ect2*, a direct target of Nkx2.5" is not referenced. Assuming that statement is from the integration of datasets used here, it hasn't been confirmed in this context. Therefore, the statement should be qualified.
6. Typo - line 117 - should be "will have an impact"
7. line 389 - The comment "accentuating the repressive effects of Nkx2.5" should be softened, as in this context it has not been shown. The data are consistent with this effect. However, the effects on DEG could be secondary.
8. Line 489 - sentence with "we exploited this amputation approach . . ." seems to be missing some words.
9. Typo - line 675 - Figure reference (Fig. B,D) is missing a number.
10. Typo - line 837 - "an" should be deleted.

11. The terms MT0, WT0, and TG0 are used in the first Figure. However, the 0 appears to actually be in reference to the timing relative to injury for comparison to the injury experiments. It isn't exactly clear to me as a reader why these terms are being used to refer to the fish prior to describing the injury experiments.

Response to Reviewers – de Sena-Tomás et al. (NCOMMS-20-47176)

We appreciate the meticulous and constructive feedback offered by the reviewers in response to our manuscript revision and are grateful for the comments describing the new version as “*substantially improved based on the presentation of new figures with much more convincing data.*” Further, the reviewers were “*enthusiastic*” and “*supportive*” that the manuscript “*will be of interest to the cardiovascular development and regeneration communities.*”

In this newly updated version, we incorporated the additional recommendations for editing and clarification of key conclusions to enhance the quality of our manuscript. Furthermore, we performed new experiments to define the detailed expression and co-localization patterns of *nkx2.5* and *ect2* at the injury border zone. Finally, below, we address each question and concern individually presented by the reviewers.

Reviewer #1

1) The manuscript would benefit from an additional round of editing for clarity. Authors are encouraged to avoid describing the study results as "exciting" "remarkable" or "innovative" as such descriptors should be left for the data to show.

We are grateful for this input and appreciate the importance of clarity in our manuscript and the benefit of utilizing more appropriate descriptors. In accordance with these suggestions, we reviewed the complete text and edited as necessary.

2) Likewise terminology such as "streamline the inquiry", "eke out" or "dampened expression" could be replaced by scientifically accurate sentences. Studies published in 2017 are referred to as "recent studies".

Similarly, we replaced the above statements with scientific terminology and refined the accuracy of the specific phrases mentioned above.

3) Nowhere in this dataset are any "direct" targets of Nkx2.5 identified. If such information has been shown in other contexts it should be made clear that it is an implication based on that other data.

We understand the importance of precision in identifying “direct” targets of Nkx2.5 and are grateful for this suggestion. Thus, we have qualified statements throughout the text where Nkx2.5 targets are discussed. Furthermore, we explicitly indicate support for our conclusion that *ect2* is putative direct target of Nkx2.5 by referring to our previously published DamID data (Ramialison et al., 2017) and additional ChIP-seq data confirming the binding of *ECT2* by NKX2-5 in human pluripotent cell-derived cardiomyocytes (Anderson et al., 2018).

Reviewer #2

1) The provided validation on sections is confusing and does not at all support the claims. psmb3, psmd7 and ect2 seem to be expressed in the scar area, not in the myocardial wound border and cannot be directly regulated by nkx2-5 based on this.

We thank the Reviewer #2 for pointing out this discrepancy as it allowed us to improve the presentation of our RNAscope data. Specifically, we performed new experiments to duplex the *nkx2.5* and *ect2* probes; these results improved our ability to illuminate co-localization of both transcripts within individual DAPI nuclei in the wound border. We have incorporated our new findings into an updated version of Fig. 7.

2) I am also worried about the lack of statistical difference in the qPCR validation.

We do appreciate the apprehension caused by our findings that only two out of 14 genes evaluated by qPCR for validation of the RNA-seq dataset achieved statistical significance. However, given the distinct nature of the techniques and variability in their sensitivities, our observation that 11 out of 14 genes illustrates directional consistency within the data is reassuring. Of course, with increased sample numbers, we would enhance our likelihood of statistical significance for more genes. However, given the challenges of generating the rescued adult mutants and the number of hearts required for each qPCR sample, we had difficulties increasing our “n”.

Reviewer #3:

1) It would appear that in total, all the data (metabolic signatures, marker analysis, etc.) support that without Nkx2.5 the adult cardiomyocytes fail to de-differentiate properly. While this would seem to be a very succinct and important message, it is not clear to me this is ever explicitly stated and the assays are always handled somewhat independently. It would be good if the authors could simply state this to unify and convey their message.

We agree that this message is succinct and important and are grateful for the input. To emphasize this conclusion, we edited statements in the Abstract and in the first and last paragraphs of the Discussion to express explicitly that Nkx2.5 is required for dedifferentiation in adult cardiomyocytes. Moreover, we also added a sentence to the Results to highlight this key finding: *“From these data, we conclude that adult CMs fail to dedifferentiate in the absence of nkx2.5 gene function.”*

2) The paragraph beginning on line 509 is a bit confusing. The stated goal is “to identify the molecular mechanisms underlying this defective reparative response.” However, most of the paragraph is setting the stage for this analysis in the next section, which is not clearly conveyed through this paragraph, and does not deal with this actual goal. I think the logic would flow better if this paragraph were revised and moved to set up the actual

comparison and defects in reparative response that is in the next section of the manuscript.

We thank Reviewer #3 for this astute feedback to improve the flow of the manuscript. In response to this recommendation, we created a distinction between the paragraph beginning at line 509 where the RNA-seq data in the injured hearts is introduced and the next paragraph where potential molecular mechanisms underlying the defective reparative response are outlined.

3) The background for the DamID data setting up the intersection of the different datasets is not covered beginning on line 624. This dataset was with mouse Nkx2.5 in HL-1 cells and is only mentioned in the Methods. The cross comparison of the data is of course fine. However, the rationale for the intersection and the background of the dataset should be provided to set up the experiment for the reader. Moreover, as the binding of Nkx2.5 and potential directness of the zebrafish targets is not confirmed in the context in the paper, statements regarding what the intersection of the data means, particularly in the discussion, need to be qualified.

We appreciate this constructive recommendation to introduce the DamID strategy more explicitly and integrate the rationale for our use of this dataset in uncovering putative targets of Nkx2.5. Furthermore, we also agree that it is important to avoid overstating the potential direct relationship of Nkx2.5 and its effectors revealed through the intersection of these datasets. Thus, we modified our conclusions and qualified statements in the Discussion, in particular.

4) There are a couple places where the references for Figure panels in the text are not in the order presented in the Figures. Fig. 2F on line 385 and Figure. 7E on line 658 are examples.

We understand this concern and the challenge that it may generate for the reader. Therefore, we reviewed all examples and changed those that were amenable to reconfiguration. However, after much consideration regarding the layout of Figure 2, we felt that any adjustments would result in other discrepancies in the flow of the manuscript text.

5) The ect2, psmb3, and psmd7 confirmation in Figure 7 show that changes in expression are primarily throughout the wound and not really in the border zone. Wouldn't the expression of these genes that are proposed to be direct targets expect to be within the de-differentiating cardiomyocytes of the wild-type heart, i.e. primarily autonomous? What does this mean? On line 647, it states, "ect2, a direct target of Nkx2.5" is not referenced. Assuming that statement is from the integration of datasets used here, it hasn't been confirmed in this context. Therefore, the statement should be qualified.

This feedback was extremely helpful in facilitating the improvement of Fig. 7 as described in our response to Reviewer #2 (point 1). As mentioned above, new duplex RNAscope data clearly demonstrate co-localization of *nkx2.5* and *ect2* transcripts in the wound border of the wild-type heart. Additionally, we mined published NKX2-5 ChIP-seq datasets in human pluripotent cell-derived cardiomyocytes and confirmed binding of NKX2-5 in the *ECT2* gene locus (Anderson et al., 2018). Altogether, these results, along with our prior study demonstrating direct binding in mouse (Ramialison et al., 2017) and our current findings depicting *ect2* misregulation upon loss of Nkx2.5 (Fig. 7), support our hypothesis that *ect2* might be directly regulated by Nkx2.5.

6) Typo – line 117 – should be “will have an impact”.

This typo has been corrected.

7) line 389 – The comment “accentuating the repressive effects of Nkx2.5” should be softened, as in this context it has not been shown. The data are consistent with this effect. However, the effects on DEG could be secondary.

This statement has been softened to separate the findings in this study and those from previous studies of the direct functions of Nkx2.5.

8) Line 489 – sentence with “we exploited this amputation approach . . .” seems to be missing some words.

We edited this sentence to fix the typo and improve clarity.

9) Typo - line 675 – Figure reference (Fig. B,D) is missing a number.

We included the figure number reference on this line.

10) Typo – line 837 – “an” should be deleted.

We removed the word “an”.

11) The terms MT0, WT0, and TG0 are used in the first Figure. However, the 0 appears to actually be in reference to the timing relative to injury for comparison to the injury experiments. It isn't exactly clear to me as a reader why these terms are being used to refer to the fish prior to describing the injury experiments.

While we do appreciate the inherent confusion in introducing the terms WT0, TG0, and MT0 prior to describing the injury experiments, we believe that establishing consistency throughout the manuscript for both the *in vivo* and bioinformatic terminology is essential. Thus, it seems wise to use a generic identifier (“0”) in the beginning of our study to set the stage for comparisons drawn later in the text between the injured heart at 7 days post

amputation (WT7) and the uninjured heart (WT0). Even though we attempted to identify a different solution regarding this nomenclature, it was difficult to find an ideal balance between these conflicting issues.

REVIEWERS' COMMENTS

Reviewer #3 (Remarks to the Author):

In the second revision of this manuscript, the authors have addressed my concerns. I do not have additional feedback to provide. I am supportive of the data and message of the manuscript.